# ATOMSURF: SURFACE REPRESENTATION FOR LEARNING ON PROTEIN STRUCTURES

Vincent Mallet[*,1,2,3,4], Souhaib Attaiki[*,1], Yangyang Miao[*,5], Bruno Correia[5], Maks Ovsjanikov[1]

* Equal Contribution
[1] LIX, Ecole Polytechnique, IPP Paris, Paris, France
[2] Mines Paris, PSL Research University, CBIO, Paris, France
[3] Institut Curie, PSL Research University, Paris, France
[4] INSERM, U900, Paris, France
[5] Institute of Bioengineering, École Polytechnique Fédérale de Lausanne, Lausanne, Switzerland
vincent.mallet@minesparis.psl.eu    maks@lix.polytechnique.fr

## ABSTRACT

While there has been significant progress in evaluating and comparing different representations for learning on protein data, the role of surface-based learning approaches remains not well-understood. In particular, there is a lack of direct and fair benchmark comparison between the best available surface-based learning methods against alternative representations such as graphs. Moreover, the few existing surface-based approaches either use surface information in isolation or, at best, perform global pooling between surface and graph-based architectures.

In this work, we fill this gap by first adapting a state-of-the-art surface encoder for protein learning tasks. We then perform a direct and fair comparison of the resulting method against alternative approaches within the Atom3D benchmark, highlighting the limitations of pure surface-based learning. Finally, we propose an integrated approach, which allows learned feature sharing between graphs and surface representations on the level of nodes and vertices *across all layers*.

We demonstrate that the resulting architecture achieves state-of-the-art results on all tasks in the Atom3D benchmark, while adhering to the strict benchmark protocol, as well as more broadly on binding site identification and binding pocket classification. Furthermore, we use coarsened surfaces and optimize our approach for efficiency, making our tool competitive in training and inference time with existing techniques. Code can be found online: github.com/Vincentx15/atomsurf

## 1 INTRODUCTION

Structural bioinformatics data is becoming available at an unprecedented pace. Advances in cryogenic Electron Microscopy (cryo-EM) in particular, have led to the production of evermore experimentally derived structures, as well as larger systems and better resolutions (Fontana et al., 2022). The development of AlphaFold (Jumper et al., 2021) along with many subsequent works have made protein structures abundantly available, with over a million high-quality predictions in the Protein Data Bank (PDB) (Berman, 2000) and over 600 million in the ESM Metagenomic Atlas (ESMatlas) (Lin et al., 2022). There is thus a growing demand for machine learning techniques which can leverage this structural data to help advance the fields of structural bioinformatics and drug design.

Protein structures are complex objects characterized both by atomic coordinates as well as intricate bio-chemical interactions between them that depend on their geometry. To be used in a learning pipeline, an initial modeling step transforming protein structures into a well-defined mathematical object is necessary. Different mathematical representations encode different structural and biological priors. For instance, the point cloud representation disregards the connectivity induced by chemical interactions, but allows for the most generic geometric description of the data. Protein *surfaces* choose to trade fine-grained information of the interior of a protein for an accurate depiction of its outer surface. This representation is thought to be of particular interest to study active interaction

sites that mostly depend on properties of the surface because of the screening effect. However, even for interactions dominated by surface terms, the knowledge of the interior can encode the stability of the surface. These different representations are illustrated in Supplementary Figure 1.

The choice of representation is particularly prominent in the context of *learning-based* methods, as specialized architectures have been developed to process each type of data. The range of approaches is studied within the field of geometric deep learning (Bronstein et al., 2017), and specialized methods have been developed to process different data types from graphs (Bruna et al., 2013; Kipf & Welling, 2016), to point clouds (Qi et al., 2017; Wang et al., 2019), surfaces (Masci et al., 2015; Monti et al., 2017), *equivariant* methods that respect a group symmetry of the data (Cohen & Welling, 2016a;b), equivariant message passing (Fuchs et al., 2020; Satorras et al., 2021) and more.

A few pioneering works have applied geometric deep learning to structural biology data representations, using 3D convolutional networks (Jiménez et al., 2017), equivariant convolutional networks (Weiler et al., 2018), sequence (Rao et al., 2021), surfaces (Gainza et al., 2020), graphs (Aumentado-Armstrong, 2018) and equivariant discrete networks (Jing et al., 2021; Stärk et al., 2022). They were followed by several others, traditionally classified based on the mathematical representation they use - see for instance Isert et al. (2023). In addition, some methods were developed ad-hock to handle protein structure, where protein properties are baked into the network (Zhang et al., 2022; Hermosilla et al., 2020; Fan et al., 2022).

In this context, the seminal work of `Atom3d` (Townshend et al., 2020) aims for a fair comparison across both different representations and learning approaches, within a well-defined protocol. Specifically the benchmark includes a set of nine benchmark tasks for three-dimensional molecular structures and establishes a consistent set of input features and parameter count to be used across all tested methods. The authors also compare different representations by evaluating neural networks based on 3D grids, graphs, and equivariant networks on the proposed tasks.

Beyond using a single representation for proteins, the simultaneous representation of a protein as several mathematical objects holds promise. Indeed, different representations encode different biological priors of the data and present different computational advantages. A well-studied combination is the use of sequence information along with a graph representation of the structure. For instance, in Hermosilla et al. (2020); Fan et al. (2022) the authors enrich the graph with additional edge types that encode the sequence. Another way to include sequence information in a graph, is to use sequence embeddings, especially ones derived from protein language models and hence benefiting from the large amounts of available sequence data (Wu et al., 2023; Zhang et al., 2023). Finally, some approaches include information derived from protein structures in the training of protein language models (Bepler & Berger, 2019; Heinzinger et al., 2023; Su et al., 2023).

## 2 MOTIVATION AND CONTRIBUTION

Despite this recent progress in comparing different representations for learning on protein data, relatively less focus has been given to *surface-based* representations, even though they have shown promising results in several applications (Gainza et al., 2020; Sverrisson et al., 2021; Wang et al., 2023). Approaches based on the surface representation have typically followed the initial `MaSIF` paper validation (Gainza et al., 2020), and hence *have never been directly compared to other representations* in the context of a single well-established benchmark. At the same time, powerful surface-based encoders have recently been proposed in the geometry processing/computer graphics literature, such as *DiffusionNet* (Sharp et al., 2022) significantly outperforming, in terms of both robustness and accuracy, the early Geodesic-CNN based techniques (Masci et al., 2015; Monti et al., 2017) which formed the basis of (Gainza et al., 2020). Unfortunately, it is not currently known how the best currently available surface-based encoders compare to other learning-based paradigms in the protein analysis tasks (e.g., on the `Atom3d` benchmark).

We fill this gap by first adapting the current state-of-the-art surface-based learning architecture to protein analysis tasks. We then perform the first fair and comprehensive comparison of a pure surface-based learning method, while adhering to the benchmark protocol, with fixed input features and parameter count. A key finding of this analysis is that surface-based encoders are competitive, but do not provide state-of-the-art results.

We then focus on exploring whether surface-based learning for protein analysis can provide *complementary* information to that of other representations. Related efforts have been made in the recent literature; in Lee et al. (2023), the authors propose to use an implicit representation of the surface as a pretraining objective. Somnath et al. (2021) proposed to first encode surface properties and use them as initial embeddings for the graph nodes, while Pegoraro et al. (2024); Xu et al. (2024) averaged the predictions made by a surface-based and a graph-based model. Nevertheless, those efforts consider surface and other (e.g., graph-based) learning *separately* and only aggregate results in a global manner (early or late fusion (Karpathy et al., 2014)). Instead, we show that by creating an integrated approach, in which the features are shared and passed between a surface and graph representation, *even within the middle layers*, allows to significantly boost performance and improve results. Crucially, by exploiting the natural spatial relations based on proximity that exist between graph nodes and surface vertices, we show that node-wise feature sharing creates a synergy between the two representations. Furthermore, we demonstrate that by using embeddings from text encoders as input features, coupled with careful and efficient architecture design, it is possible to achieve unprecedented state-of-the-art results on a wide range of tasks.

To summarize, our key contributions include:

- An adapted design of the recent state-of-the-art *DiffusionNet* architecture, which addresses some of its limitations (including instabilities and scale independence) in the context of protein analysis tasks.
- The first comprehensive comparison of surface-based learning against alternative representations such as graphs or grids within an established benchmark.
- A novel integrated approach, which is based on node-wise feature-sharing across all learned layers, between surface and graph-based encoders that are learned jointly.
- State-of-the-art results in a wide variety of challenging scenarios and enhanced computational throughput by using residue graphs and coarsened meshes.

The rest of the paper is organized as follows: in Section 3.1, we present the surface representation we use and the specialized networks used to process it. Section 3.2 highlights a challenge for surface networks learning on various scales and presents solutions to mitigate these issues. In Section 3.3, we propose to synergistically integrate graph and surface information within a unified architecture, harnessing the power of both representations. We provide the details regarding the chosen architecture in Section 3.4 and analyze its computational aspects in Section 3.5.

## 3 METHODS

### 3.1 SURFACE REPRESENTATION LEARNING

Our first objective is to study the utility of the best existing surface encoders for learning on protein data. To generate the surface representation $\mathcal{S}_{\mathbf{P}}$ of the protein $\mathbf{P}$, we rely on MSMS and on mesh coarsening and cleaning steps, detailed in Appendix C.1. At the basis for our surface-based learning method, we then employ the *DiffusionNet* approach (Sharp et al., 2022). This method has been proven to be highly robust and effective across a diverse set shape analysis tasks (e.g., (Attaiki et al., 2021; Cao & Bernard, 2022; Sun et al., 2023; Li et al., 2022) among others). In particular, *DiffusionNet* avoids using local patch parametrizations, which can lead to instabilities across different mesh structures and enables *long range and multi-scale information* propagation by using learned diffusion for information sharing. The mathematical foundation of *DiffusionNet* is the heat equation, which simulates the diffusion of heat, or, equivalently, the behavior of Brownian motion on a surface over time. For a surface $S$, let $\mathbf{f}_t : S \to \mathbb{R}$ be the function that defines the heat distribution over $S$ at time $t$ and $\Delta_S$ the Laplace-Beltrami operator (Meyer et al., 2003; Vallet & Levy, 2008) of the surface. The heat equation is the linear differential equation below (equation 1), solved by diagonalizing the Laplace-Beltrami. In practice, we store the $k = 128$ smallest eigenvalues of $\Delta_S$ in a diagonal matrix $\Lambda \in \mathbb{R}^{k \times k}$, with the corresponding eigenvectors in $\Phi \in \mathbb{R}^{n \times k}$, and vertex area weights $M \in \mathbb{R}^{n \times n}$. The spectrally truncated solution to the heat equation is given as $\mathbf{f}_t = \Phi e^{-\Lambda t}(\Phi^T M)\mathbf{f}_0$.

$$\text{(Heat equation)} \qquad \frac{\partial \mathbf{f}}{\partial t} = \Delta_S \mathbf{f}. \qquad (1)$$

In *DiffusionNet*, this equation is used to perform information propagation on a surface using a feature map as $\mathbf{f}_0$ and *learning* the diffusion time in a task-specific manner. This mechanism relies on dense linear algebra operations, offering straightforward differentiation with respect to both $\mathbf{f}$ and $t$. The diffusion layers are combined with features based on $\mathbf{f}_t$, their spatial gradients and standard, point-wise MLPs. This leads to an architecture which can capture multi-scale geometric details of a surface in a task-specific manner.

## 3.2 Adapting to Protein Surfaces of Diverse Scales

As we demonstrate in Section 4 below, our first empirical observation is that applying the *DiffusionNet* architecture directly to protein datasets, without modifications, yields relatively poor performance. We attribute this primarily to the fact that the initial *DiffusionNet* architecture targeted applications involving related near-isometric shapes (e.g., humans in different poses). A key technical issue is that most existing approaches using *DiffusionNet*'s normalize all shapes to a uniform surface area. This step ensures that shapes have the *same global scale* which stabilizes learning. Unfortunately, in the context of protein analysis, tasks such as ligand-binding preference determination depend critically on the relative sizes of proteins and ligands, making considerations on the *scale* of proteins essential, and global scale normalization would lose this precious information.

On the other hand, the efficacy of *DiffusionNet*'s *receptive field* is contingent upon the diffusion times learned within each diffusion layer. Without scale normalization variations in the sizes input shapes can lead to discrepancies in the network's learned receptive field. This is elucidated by the following well-known proposition, whose proof is provided in the supplementary material for completeness:

**Proposition 3.1.** *Let $X$ be a shape and $Y = \alpha X$ its scaled version by a factor $\alpha > 0$. Denoting by $E_{\cdot}(t, x)$ the expected geodesic distance for a Brownian motion starting from point $x$ after time $t$, it holds that: $E_Y(t, x) = \alpha E_X\left(\frac{t}{\alpha^2}, x\right)$.*

Importantly, this result suggests that the *time parameter* of diffusion must be *adapted* depending on the scale, whereas in *DiffusionNet* the learned time parameters are *shape independent*. We also remark that in scenarios involving non-isometric surfaces, such as proteins that might have different scales, learned diffusion, especially if it is learned without biological considerations, can generalize poorly across highly diverse shapes, and lead to instabilities during training.

To address this issue, we enhance the original *DiffusionNet* framework in two ways. First, we enable support for batch (the original model was limited to batch sizes of one) and incorporate a Batch Normalization layer (Ioffe & Szegedy, 2015) after each diffusion layer to stabilize learning. Second, we facilitate the optimization process by incorporating *biological priors* relevant to spatial scales. Consequently, we determined that diffusion times around 10 resulted in receptive fields around 10 Å (see Supplementary Figure 2), which aligns with the spatial scale of binding sites. Inspired by the inherent multi-scale nature of protein structures, we opted to draw samples from a normal distribution $t \sim \mathcal{N}(10, 5)$, characterized by a relatively high variance. The absolute values of these samples were then utilized as the initial values for our diffusion timescales. Both the large scale and the large variance are retained during training, as illustrated in Supplementary Figure 3), enabling efficient multi-scale and long-distance message passing. These enhancements to *DiffusionNet* implementation mitigate the instabilities in the training process (as seen in Supplementary Figure 2) and are available in the provided code repository and as a `pip` package.

## 3.3 Hybrid Representation Learning

As mentioned earlier, beyond assessing the efficacy of surface-based learning compared to other representations for protein learning, we explore the benefits of integrating different representations in a unified framework, harnessing their distinct strengths. Intuitively, surface representations can capture the intricate geometric details critical for tasks involving protein interactions, while graph representations detail the specific atomic *interactions* within a protein's interior that indirectly influence its surface dynamics and interaction capabilities. Furthermore, these representations facilitate complementary approaches to learning: local message passing through graphs and global information dissemination for surfaces via learned diffusion. Inspired by these considerations, we propose a method that enables feature sharing *between* graph and surface-based representations. As

emphasized below, unlike previous related approaches (Somnath et al., 2021; Pegoraro et al., 2024), we enable communication between the two representations across *all* learned layers of the network.

As a basis for our hybrid representation, in addition to the surface $\mathcal{S}_{\mathbf{P}}$, we construct a graph representation $\mathcal{G}_{\mathbf{P}} = (\mathcal{V}_g, \mathcal{E}_g)$. We use either a graph whose nodes are atoms, which aligns with the one used within the `Atom3d` benchmark, or one defined at the residue level. The residue-level graph is enriched with ESM-650M (Rives et al., 2021) sequence embeddings used as node features.

To construct our hybrid approach, we then build a bipartite graph $G = (V, E)$, where $V = \mathcal{V}_g \cup \mathcal{V}_s$ represents graph nodes and surface vertices, respectively. For each vertex on the surface, we find its 16 nearest neighbors in the graph and add the corresponding bidirectional edges in the bipartite graph. We provide a more detailed description of the construction and features of the atomic, residue and bipartite graphs in Appendix C.1.

We now define block operations to encode a protein using $\mathcal{S}_{\mathbf{P}}, \mathcal{G}_{\mathbf{P}}$ and $G$. Denote encoders on surfaces and graphs as $s_\theta$ and $g_\theta$, respectively, and the set of input features as $\mathcal{X} = \{x_n, n \in \mathcal{V}\}$. The corresponding encoded features are $\mathcal{H} = \{h_n, n \in \mathcal{V}\}$ with $h_n = s_\theta(x_n)$ for nodes $n \in \mathcal{V}s$ and $h_n = g_\theta(x_n)$ for nodes $n \in \mathcal{V}_g$. Our general methodology incorporates message-passing neural networks, denoted $\text{MP}_\theta$, over the bipartite graph $G$, such that at a layer $l$, we get $\mathcal{X}^{l+1} = \text{MP}_\theta^l(\mathcal{H}^l)$. By employing distinct sets, $\theta_{sg}$ and $\theta_{gs}$, the architecture handles messages traversing from the surface to the graph and vice versa. Those block operations can be stacked as shown in Figure 1. We emphasize that our feature sharing occurs on the local (node) level and is enabled by the proximity relations in 3D space. Moreover, our hybrid approach trains both representations *jointly*, while enabling information sharing across *all network layers*, which is crucial to its success.

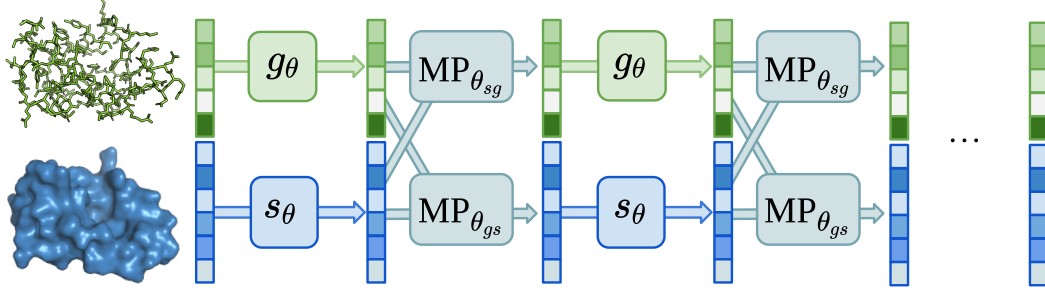

Figure 1: Illustration of our approach integrating surface and graph information. We ensure *joint learning* across the two representations and enable information propagation across *all* layers of the network. Our information sharing is based on the spatial proximity relations between individual graph nodes and surface vertices (not shown here).

## 3.4 PROPOSED ARCHITECTURES

Our framework incorporates surface encoding blocks, $s_\theta$, which consist of a diffusion operation followed by a pointwise neural network with two hidden layers of a specified width. The first network we propose, `Surface Diff` is only based only on those surface blocks. It is used to assess the relevance of the surface representation used in isolation. Note that `Surface Diff` is based on *DiffusionNet* but incorporates our modifications mentioned in Section 3.2. For all other methods that use a hybrid approach, the widths for both encoding blocks $s_\theta$ and $g_\theta$ were consistently set to equal values.

We introduce `AtomSurf-bench`, a model aimed at comparing representations in a fair way by following the `Atom3d` benchmark protocol. Its graph encoder $g_\theta$ consists of Graph Convolutional Networks (GCN) (Kipf & Welling, 2016), intertwined with Batch Normalization operations and its message-passing over the bipartite graph is a Graph Attention Layer (Veličković et al., 2017). Following the `Atom3d` benchmark standards, `AtomSurf-bench` has 200k learnable parameters and does not use surface input features, only considering the atom type onto its atomic-level graph representation.

In addition, we introduce `AtomSurf` that makes use of the aforementioned residue graph, along with the recently-proposed ProNet (Wang et al., 2022a) encoder. Despite acting on residue graphs, ProNet adds a featurization of the geometric conformation of the atoms composing each residue,

based on relative local coordinate systems that uniquely and equivariantly identify the conformation, allowing for completeness (as introduced in ComeNet (Wang et al., 2022b)). Those features are then embed using spherical harmonics, and ProNet relies on the message passing introduced in GraphConv (Morris et al., 2019) to perform learning. In addition, we perform our message passing using the aforementioned bipartite graph features and a GVP (Jing et al., 2021) encoder.

Different bipartite message passing networks and organization are possible, encompassing as special cases several existing approaches. We provide the implementation details in Appendix C. We evaluate these configurations and conduct several ablation studies on our final model, in Section 4.5.

### 3.5 COMPUTATIONAL ENHANCEMENTS

Surface-methods are traditionally thought to be compute-expensive methods, motivating approaches to side-step their intrinsic complexity (Sverrisson et al., 2021). Through a complexity analysis presented in Appendix D.1, we found the number of vertices to be critical in *DiffusionNet* runtime, which we addressed by coarsening our meshes. In this coarse surface regimen, the graph encoding with ProNet is the computational bottleneck, which is smaller than when used in isolation. Hence, we claim that our method is faster than this efficient graph encoder in the coarse surface regimen.

From the storage and memory perspectives however, the memory footprint of the surface-related operations dominate the ones originating from the graphs, especially because of stochasticity. We did not find it to be limiting in terms of I/O, but rather in terms of batch size. We implemented a dynamic batching procedure, alleviating our memory issues. Further details are provided in Appendix D.2. Finding ways to reduce the memory footprint of surface networks remains an important direction.

## 4 RESULTS

### 4.1 PERFORMANCE OF SURFACE REPRESENTATION

We start by validating our surface encoder on the RNA segmentation benchmark (Poulenard et al., 2019) for surface methods. The task is to segment 5s ribosomal RNA molecules into functional components. This dataset consists of 640 RNA surface meshes of about 15k vertices, and was already used to compare modern surface encoders. We assess the impact of the proposed enhancements to *DiffusionNet* by showing the learning curves of the enhanced models on the RNA segmentation task (see Figure 2 and Appendix F.1 for a similar analysis on PSR). Moreover, we compare their performance to other recent surface encoders, DGCNN (Wang et al., 2019) and DeltaConv (Wiersma et al., 2022) and report results in Table 1. Our experiments show that our adjustments significantly improve the *DiffusionNet* stability issues, and allow it to converge to a model with a test accuracy of 84.1% instead of of 80.9%. This accuracy is significantly higher than DGCNN (74.7%) and DeltaConv (78%).

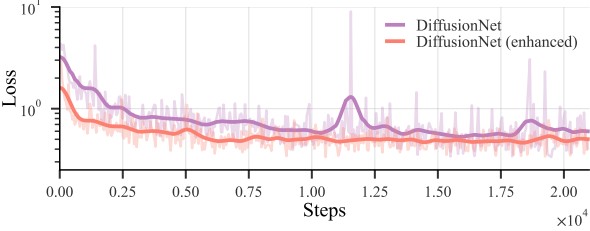

| Method | Accuracy |
|---|---|
| DGCNN | 74.7 |
| DeltaConv | 78 |
| DiffusionNet (original) | 80.9 |
| DiffusionNet (enhanced) | **84.1** |

Figure 2: Learning curve on the RNA segmentation using the original and our enhanced DiffusionNet models.

Table 1: Performance of Different Surface Encoders on the RNA Segmentation Task.

Then, to compare the performance of our surface encoder to the use of other representations, we turn to the Atom3d benchmark, focusing on its three tasks exclusive to proteins. The Protein Interaction Prediction (PIP) task aims to predict which part of a protein interacts with another, and holds about 120k examples. The Mutation Stability Prediction (MSP) task is to determine if a mutation enhances the stability of protein-protein interaction (5k examples). Finally, the Protein Structure Ranking (PSR) task aims to assign a quality score to predicted protein structures with around 10k proteins.

A more detailed description of all our datasets is available in Appendix E.

We anticipate that the surface representation will excel in tasks involving interactions, like PIP, but may underperform in the PSR task, which is heavily influenced by subtle internal changes in the protein volume that might not affect the surface. The results are presented in Table 2.

Surprisingly, we observe that the surface method `Surface Diff`, despite using a state-of-the-art surface encoder, consistently falls short in its performance, even on the protein-protein interaction task. Such an observation challenges assertions in previous purely surface-based methods, and highlights the importance of direct benchmarking in general. Despite promising modeling of protein interfaces, which are intrinsically surface objects, the network could not reach satisfactory test performance. We emphasize that all networks are trained in a vanilla setting, in particular, unlike `MaSIF`, our input features are minimalistic. Surface networks may excel when supplemented with richer information. However, when all input features are held constant, surface networks do not emerge as top performers.

Table 2: Comparison of different representations, including surface performance. Dashes in the equivariant methods' column indicate that these methods could not be used due to memory constraints.

|  | PIP | MSP | PSR | |
| --- | --- | --- | --- | --- |
|  | Auroc | Auroc | $R_l$ | $R_g$ |
| 3DCNN | 0.844 | 0.574 | 0.431 | 0.789 |
| ENN | - | 0.574 | - | - |
| Graph | 0.669 | 0.609 | 0.411 | 0.75 |
| Surface Diff | 0.837 | 0.5 | 0.33 | 0.643 |
| AtomSurf-bench | **0.876** | **0.707** | **0.452** | **0.831** |

### 4.2 Synergy in Combined Representations

In this section, we assess the performance of our proposed hybrid methods, which have the particularity of combining surface and graph representations. First, we use the same experimental setup and compare our `AtomSurf-bench` to the previous models exclusively grounded in one representation. Our results are reported in the last row of Table 2.

Our method outperforms single modalities in all three tasks, while adhering to the `Atom3d` benchmark protocol. This result highlights the synergy between the two representations. Achieving this is noteworthy because, first, the input features remain minimal, and second, to maintain a consistent parameter count, both the graph and surface encoders are considerably condensed. An interesting result is that even in the case of PSR, where surfaces do not intuitively seem relevant, the mixed model outperforms its graph counterpart with a comfortable margin. One possible interpretation for this result is *DiffusionNet*'s ability to perform long-range message passing.

In addition, we compare to popular, state-of-the-art models outside of the benchmark constraints. Namely, we compare against ProNet (Wang et al., 2022a) and GearNet (Zhang et al., 2023), which are graph-based methods at the residue level, but encode the local geometry of residues as features; and to the atomic-level extension of GVP (Jing et al., 2021). Finally, we include their extension with features derived from protein language models (Zhang et al., 2023). Please note that differing parameters and training procedures do not necessarily follow the protocol of the original benchmark. For this reason, we also include the performance of our second proposed model, `AtomSurf`, which uses all input features and more recent encoders. Out of fairness, we follow exactly the training protocol and problem formulation for each considered task, and use externally reported performance for all other methods. The results are presented in Table 3.

Our first result is that even `AtomSurf-bench` is competitive with more highly engineered approaches and outperforms them in two of the three evaluated tasks. It even outperforms pretrained methods on the PIP task. The PSR task, which is aimed at detecting non-canonical protein structures, appears to benefit methods that integrate *explicit* biological insights into protein geometry, such as the computation of side-chain angles.

Moreover, `AtomSurf` successfully results in another performance boost on this benchmark, setting a new clear state-of-the-art performance on the PIP and MSP tasks, and closing the gap on PSR. Importantly, our method does not rely on pretraining and uses less than half the number of parameters than any other method.

Table 3: Performance of our hybrid approach on benchmark tasks, compared to state-of-the-art models. Best result in bold, second best underlined.

| | PLM | Params | PIP Auroc | MSP Auroc | PSR $R_l$ | PSR $R_g$ |
|---|---|---|---|---|---|---|
| ProNet | | 2M | 87.1 | - | **63.2** | 84.9 |
| GVP | | 11M | 86.6 | 68.0 | 51.1 | 84.5 |
|    *+ pretraining* | | 11M | 87.4 | 71.1 | 51.5 | 84.8 |
| GVP-ESM | ✓ | 11M | - | 61.7 | - | **86.6** |
| GearNet-ESM | ✓ | 42M | - | 68.5 | - | 82.9 |
|    *+ pretraining* | ✓ | 42M | - | 70.2 | - | 86.3 |
| AtomSurf-bench | | 200k | 87.6 | 70.7 | 45.2 | 83.1 |
| AtomSurf | ✓ | 600k | **90.9** | **71.6** | 61.7 | 85.7 |

## 4.3 EVALUATION ON PROTEIN INTERACTION PREDICTIONS

In addition to the `Atom3d` benchmark tasks, we evaluate our approach on the task of protein binding sites prediction. Binding sites are regions of protein surfaces where a partner (that can be another protein, an RNA molecule, a small molecule) interacts with the protein. A fine characterization of those regions helps in understanding protein functions and plays a significant role in drug design.

**Masif-ligand** We start by evaluating our proposed approach on the task of ligand-binding preference prediction for protein binding sites, introduced in (Gainza et al., 2020). It holds 1459 protein structures bound to one of the seven most common ligands in the PDB. Given a binding site, the task amounts to predicting its corresponding co-factor. We benchmark our approach against MaSIF (Gainza et al., 2020) and HMR (Wang et al., 2023), with the latter being recognized as state-of-the-art on this task. Additionally, considering the notable performance of ProNet on the previous benchmark, we include it in this experiment. We use balanced accuracy (the average recall achieved for each class) as our performance metric consistent with prior studies.

As depicted in Table 4, our method surpasses the existing methods and sets a new state-of-the-art for this task. ProNet achieves a disappointing AuROC of 0.75. Unlike HMR, which relies solely on surface representation, our results further validate the efficacy of synergistically combining surface and graph representations, showcasing that this integration leads to superior performance in ligand-binding pocket classification.

**Antibody epitope prediction** Improved understanding and ability to predict the interaction between an antibody and its target, denoted as the antigen, has direct applications in antibody-based treatments, which represent a highly promising therapeutic avenue (Kaplon et al., 2023; Jamali et al., 2024). Pegoraro et al. (2024) introduced a dataset containing 235 antibody–antigen complexes. The task consists in predicting interacting residue on the antibody and antigen, from their two structures. The method proposed in their article, denoted as `GEP`, is a relevant comparison to ours, because it averages predictions made by a surface model and by a graph-based model. We present the results in Figure 3.

| | AuROC |
|---|---|
| MaSIF (Gainza et al., 2020) | 0.74 |
| Pronet (Wang et al., 2022a) | 0.75 |
| HMR (Wang et al., 2023) | 0.81 |
| AtomSurf-bench | 0.84 |
| AtomSurf | **0.88** |

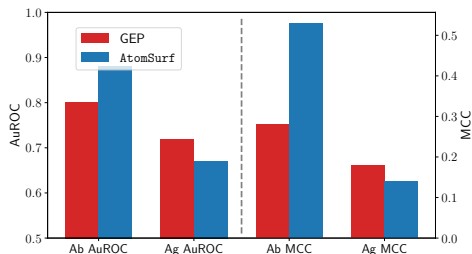

Table 4: Balanced accuracy of our hybrid approach on the *MaSIF-ligand* task.

Figure 3: Performance of our model on the binding site detection task.

As can be seen in our results, `AtomSurf` has a comparable performance to `GEP` on the antigen binding site prediction, but outperforms it with a comfortable margin on the antibody side (+0.25 MCC points). An accurate prediction of the antibody residues involved in the antigen recognition is key for antibody specificity optimization.

**Validation on PINDER**  In addition to this relatively small dataset, we validate our approach on the recently proposed, large-scale PINDER dataset (Kovtun et al., 2024). We used the clustered version of this dataset that holds around 42k structures. Systems are split rigorously, based on interface similarity. Moreover, test systems are also available in their unbound form (Apo setting) and as predicted by AlphaFold2 (AF2 setting), representing a more realistic use case. We consider a first task formulated as PIP (*Pinder-Pair*) of predicting interacting pairs of residues and another close to Masif-Site (Gainza et al., 2020), only taking one protein as input to predict interacting residues (*Pinder-site*). We compare to ProNet in Table 5, and include a comparison of accuracies in Supplementary Table 3.

Table 5: Auroc of our method compared to ProNet on the PINDER dataset.

| Task | *Pinder-Pair* | | | *Pinder-Site* | | |
|---|---|---|---|---|---|---|
| Split | Holo | Apo | AF2 | Holo | Apo | AF2 |
| Pronet | 80.1 | 78.2 | 73.5 | 74.3 | 70.7 | 60.6 |
| AtomSurf | **92.8** | **88.4** | **87.1** | **88.3** | **84.2** | **82** |

`AtomSurf` widely outperform the ProNet baseline on all tasks, splits and metrics. The performance over the *apo* set is decreased compared to the *holo* set, as well as the performance on predicted structures, as expected (Huang et al., 2024). However, our network is able to retain a remarkable accuracy across the board, validating its robustness.

## 4.4 ADDITIONAL RESULTS

**Visualization of our predictions**  We display the interaction probability predicted by our model across two protein surfaces, a homodimer and a heterodimer, and plot the results in Figure 4. Despite minor prediction errors, such as the misidentification of residues in the lower region of `3dbh`, our results clearly show that the model effectively identifies binding sites on proteins.

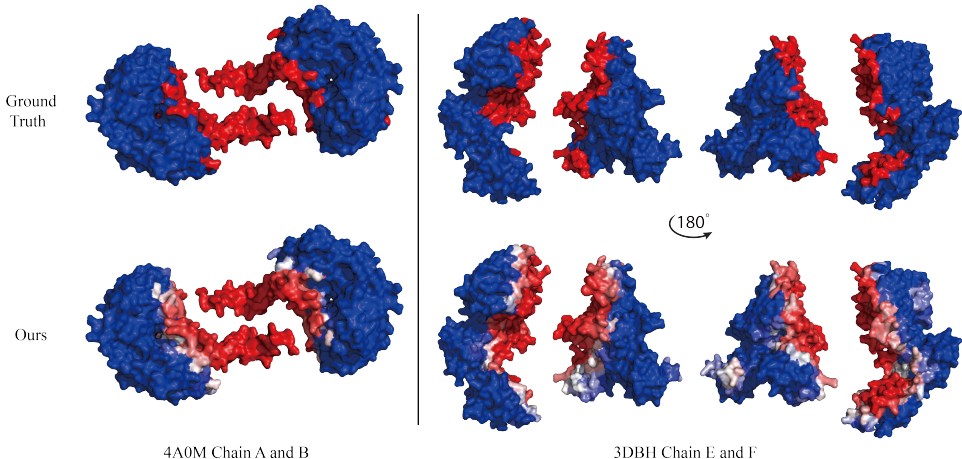

Figure 4: A qualitative view of our results: The top row shows the ground truth with interaction sites marked in red, while the bottom row displays our predictions. On the left, the interaction between chains A and B of the system with PDB ID `4A0M` is illustrated. The two rightmost columns depict chains E and F of the protein with PDB ID `3DBH` from two rotated perspectives of 180°.

**Learned time analysis**  In the Supplementary Fig. 2 and 3, we provide visualizations showing both the relation between the effective receptive field and diffusion times, as well as the distribution of

learned times on the MSP task. These visualizations demonstrate that long-range communication enabled by learned diffusion is indeed useful, and can contribute to overall performance, thus shedding light on the relative strengths of surface-based learning.

**Assessment of the reproducibility** Supplementary Table 2 shows the standard error of our estimate of the mean performance of `AtomSurf`, assessing its reproducibility and the significance of its difference. Across tasks, the performance gap is higher than the standard error.

## 4.5 ABLATION STUDY

**Model ablations** Finally, we examine the impact of different design choices on our tasks. In Appendix C, we introduce the `sequential` and `bipartite` scenario, which amount to variations in the connectivity of the blocks. Another major design choice is the choice of the Message Passing (MP) component. We explore several options, including discarding the geometric notion of a neighborhood, allowing for potentially long-distance message passing (`Att.` setting), as well as varying the number of blocks. Our detailed results are presented in Appendix F.4.

The `sequential` strategy displays underwhelming results (70.7 vs 60.9 on MSP for example), which could root from its incapacity to handle multi-scale, simultaneous message passing. Among the `bipartite` settings, `Att.` is consistently outperformed by the localized message-passing networks. In short, the best setting leverages several blocks of message passing, with enhanced results for the most interconnected networks. In addition, the geometry of the bipartite graph is important, and an attentive mechanism generally benefits the optimal mixing.

**ESM embeddings ablation** Due to the strong performance of protein language models, we also perform an ablation by removing the ESM embeddings from our graph initial node features. Ablated model's performance decreases as expected, showing that ESM is truly an important feature. However, even without ESM, our method retains state-of-the-art performance on most tasks, notably on interaction tasks such as Masif-Ligand (84 vs 81), PINDER (overall +10 AuROC points) and others (detailed results in Appendix F.5). Interestingly, on the PINDER dataset that has stricter splits, the performance drop seems to be reduced. One possible explanation for this result would be that those sequence-derived features contribute to memorization, whereas learned structural properties lead to better generalization. We believe that a more in-depth investigation of this phenomenon is an interesting direction for future work.

## 5 CONCLUSION & LIMITATIONS

In this paper, we analyzed the utility of the surface representation in machine learning on protein structures. We first adapted the design of the recent *DiffusionNet* in the context of protein analysis tasks, and compared it against other methods by adhering to the `Atom3d` benchmark protocol, revealing both the promise and the limitations of surface-only learning. We then introduced a novel integrated architecture that combines surface and graph representations and achieves state-of-the-art results across several benchmark tasks. Key to our approach is a node-wise information sharing mechanism, which allows for joint training of graph and surface representations, coupled with *localized information propagation* across all network layers. Our ablation analysis further highlights that simplistic approaches to combining different representations lead to suboptimal performance. We also demonstrated the performance of our approach in identifying antibody-antigen and ligand-binding preferences achieving state-of-the-art results.

Our work strongly supports the notion that leveraging *multiple* representations with unique strengths is a promising strategy for advancing protein analysis. Moreover, integrating biological priors both within the learning frameworks and into information sharing across representations seems crucial for achieving high accuracy in challenging scenarios. A validation on real-life scenarios is our next step to fully establish this method. Despite our optimizations, one of the current limitations of our approach is its significant memory requirements, highlighting the need for computationally more efficient surface-based pipelines, in line with suggestions by Sverrisson et al. (2021). Beyond addressing these challenges, our work can help pave the way to more powerful integrated multi-modal solutions for additional tasks within structural bioinformatics, including *generative* modeling, but also investigating ways in which information sharing across both *representations* and *tasks* can lead to improvements in robustness and accuracy.

## ACKNOWLEDGEMENTS & FUNDING

The authors would like to thank Emanuele Rodolà and Marco Pegoraro for discussions that contributed to initiate this project. They also would like to thanks Erkan Turan for trying to setup more tasks, as well as Victor Laigle for making the Supplementary Figures on batch size variability.

V.M., S.A. and M.O. are supported by the ANR Chair AIGRETTE, the ERC Starting Grant No. 758800 (EXPROTEA) and the ERC Consolidator Grant No. 101087347 (VEGA). V.M. was additionally supported by DataIA and Sanofi. Y.M and B.C. are supported by Swiss National Science Foundation grants 310030_197724, TMGC-3_213750 and 200020_214843. This work was performed using HPC resources from GENCI–IDRIS (Grant 2023-AD010613356) and CITAS at EPFL.

## BROADER IMPACT

This paper presents work to advance the field of machine learning on protein structure. There is no direct ethical or societal implication of this work, but potential indirect ones, none of which we feel must be specifically highlighted here.

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

APPENDIX

In this document, we compile all results and discussions that could not be included in the main manuscript due to page constraints.

More specifically, we first provide additional data illustrations in Appendix A. Then, we provide a proof for Proposition 3.1 in Appendix B. Following this, we detail the specifics of our implementation in Appendix C. Then, we provide an analysis of the computational aspects of our technique in Appendix D. Finally, we provide additional results in Appendix F.

## A  ADDITIONAL DATA ANALYSIS

In Supplementary Figure 1, we illustrate the different representations that exist for protein structures.

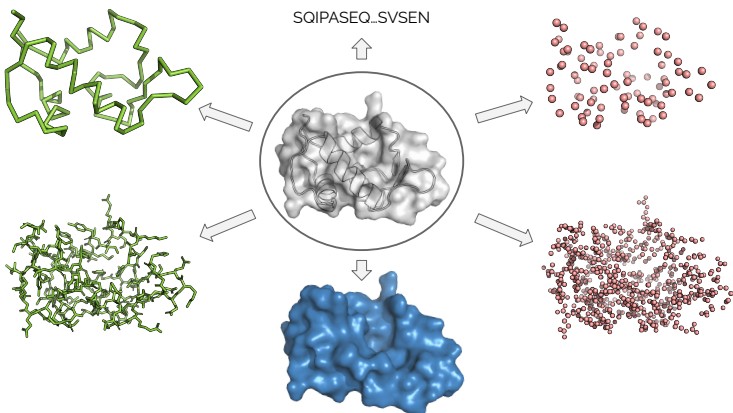

Supplementary Figure 1: Diverse mathematical objects used to represent a protein structure, sequences, molecular surfaces (blue), atom-level and residue-level point clouds (red) and graphs (green). Effective machine learning for protein structures hinges on selecting the appropriate mathematical representation along by a compatible machine-learning technique.

### A.1  ASSESSMENT OF THE SCALE TO USE

In Figure 2, we plot the results of diffusing a Dirac initial distribution for several diffusion times, over the surface of the MDM2 protein, involved in apoptosis and cancer treatment. On the figure we see its groove, which corresponds to its binding site. Orange vertices correspond to the ones that have the highest probabilities, such that their sum represents 90% of the probability mass. A diffusion time of ten seems to convey a relevant scale for this binding site.

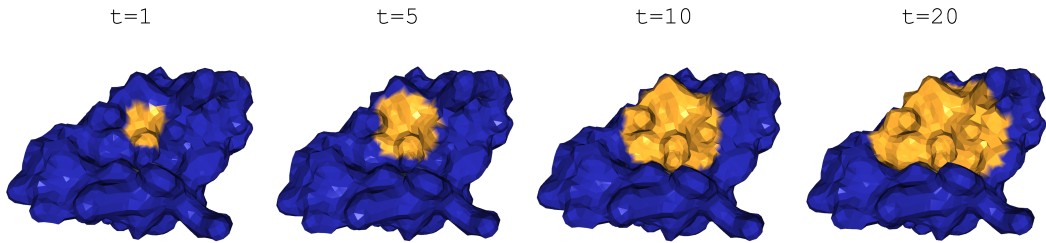

Supplementary Figure 2: Visualization of a Dirac delta function $\delta_x$ at a point, diffused for several diffusion times.

In Figure 3, we plot the diffusion times learnt by a network (on the MSP task). We see that the network keeps a variety of scales, and has learnt to use large scales, effectively enabling long-distance message passing.

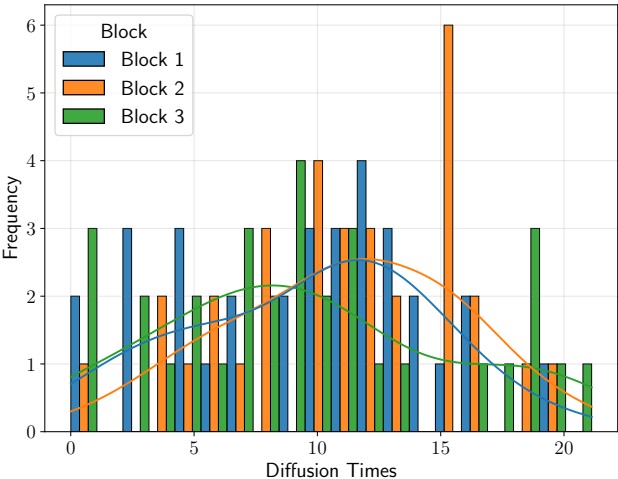

Supplementary Figure 3: Histogram of the diffusion times obtained after training.

## B  PROOF OF PROPOSITION 3.1

*Proof.* For simplicity we use the computations in the discrete setting (on meshes). Everything remains the same in the smooth (surface) setting, however. Let $M_1$ represent a shape modeled as a triangular mesh, with $W_1$ and $A_1$ denoting its cotangent Laplacian and area matrices, respectively. Therefore, its eigenvalue decomposition satisfies:

$$W_1\phi^1 = \lambda^1 A_1 \phi^1.$$

Suppose $M_2$ is another mesh that is a scaled version of $M_1$ by a scaling factor $a$. Then, according to (Bronstein & Kokkinos, 2010):

$$A^2 = a^2 A^1$$
$$\lambda^2 = \lambda^1/a^2$$
$$\phi^2 = \phi^1/a$$

The heat kernel can be computed as per (Sun et al., 2009):

$$k_t(x,y) = \sum_i \exp(-\lambda_i t)\phi_i(x)\phi_i(y).$$

Consequently, we have:

$$
\begin{aligned}
k_t^2(x,y) &= \sum_i \exp(-\lambda_i^2 t)\phi_i^2(x)\phi_j^2(y) \\
&= \sum_i \exp(-\lambda_i^1/a^2 t)\phi_i^1(x)\phi_j^1(y)/a^2 \\
&= k_{t/a^2}^1(x,y)/a^2
\end{aligned}
$$

If $g(x,y)$ denotes the geodesic distance between two points on the surface $x$ and $y$, then:

$$g^2(x,y) = a g^1(x,y).$$

Denoting $E_{.}(t, x)$ as the *expected geodesic distance* of a Brownian motion starting from $x$ after time $t$, we find:

$$
\begin{aligned}
E^{\mathbf{2}}(t, x) &= \sum_y k_t^{\mathbf{2}}(x, y) g^{\mathbf{2}}(x, y) A^{\mathbf{2}}(y) = \sum_y k_t^{\mathbf{2}}(x, y) a g^{\mathbf{1}}(x, y) a^2 A^{\mathbf{1}}(y) \\
&= \sum_y k_t^{\mathbf{2}}(x, y) a g^{\mathbf{1}}(x, y) a^2 A^{\mathbf{1}}(y) \\
&= \sum_y (k_{t/a^2}^{\mathbf{1}}(x, y)/a^2) a g^{\mathbf{1}}(x, y) a^2 A^{\mathbf{1}}(y) \\
&= a \sum_y k_{t/a^2}^{\mathbf{1}}(x, y) g^{\mathbf{1}}(x, y) A^{\mathbf{1}}(y) \\
&= a E^{\mathbf{1}}(t/a^2, x).
\end{aligned}
$$

Interestingly, if we assume $E(t, x) = \sqrt{t}$ (as in the Euclidean setting), then: $a E^{\mathbf{1}}(t/a^2, x) = a\sqrt{t/a^2} = \sqrt{t} = E^{\mathbf{2}}(t, x)$. □

## C   IMPLEMENTATION DETAILS

### C.1   PROTEIN REPRESENTATION DETAILS

**Surface representation**   To generate the surface representation $\mathcal{S}_{\mathbf{P}}$ of the protein $\mathbf{P}$, our initial step involves computing the protein surface using MSMS (Sanner et al., 1996). The resulting meshes are usually quite large, but in the rare cases of small proteins, we incrementally increase sampling density until we achieve a minimum number of 256 vertices. Then, we employ quadratic decimation (Garland & Heckbert, 1997), to embed our surfaces into coarser and more compact meshes, again ensuring a minimum vertex count. Coarsening the mesh has an impact on computational efficiency and on performance, assessed in Sections 3.5 and 4. Finally, we ensure mesh quality by removing non-manifold edges, duplicated or degenerate vertices and faces, as well as small disconnected components. We compute several surface features: the Gaussian and mean curvature, as well as the shape index at each vertex. We also include the normal as well as the heat kernel signature (Sun et al., 2009) at each point.

**Graph representation**   In addition to the surface $\mathcal{S}_{\mathbf{P}}$, we construct a graph representation $\mathcal{G}_{\mathbf{P}} = (\mathcal{V}_g, \mathcal{E}_g)$. To ensure consistency and fairness in comparison within the `Atom3d` benchmark, we follow their conventions: each node in the graph corresponds to an atom within the protein and edges $\mathcal{E}_g$ are defined between pairs of atoms that are within a 4.5 angstrom radius cutoff. The only initial node features are one-hot encoding of atom types.

Independently, we construct another graph representation to be used outside of the benchmark setting. We choose to use residue graphs for computational efficiency: each node in this graph represents an amino-acid residue and edges are introduced between pairs of atoms that are within a 12 Å radius cutoff. For each residue, we add a one hot encoding of its type, its secondary structure annotation and its hydrophobicity. We also compute sequence embeddings using the ESM-650M (Rives et al., 2021) pretrained model and include them as node features.

**Bipartite graph construction**   To construct our hybrid approach, we start by building a bipartite graph $G = (V, E)$, where $V = \mathcal{V}_g \cup \mathcal{V}_s$ represents graph nodes and surface vertices, respectively. For each vertex on the surface, we find its 16 nearest neighbors in the graph and add the corresponding bidirectional edges in the bipartite graph. We also experimented with edges based on a geometric neighborhood of 8 Å, corresponding to the value widely used to define contact maps (Fariselli et al., 2001). However, distance-based cutoffs result in a varying number of neighbors which we found to make learning less stable.

Outside of the benchmark setting, we add edge features to our bipartite graph. Given an edge $e_{g,s}$ tying a graph node to a surface vertex with normal $\vec{n}_s$, we use its direction $\vec{u}_{g,s} = \frac{x_s - x_g}{||x_s - x_g||}$ as a

vector edge feature. The associated distance and angle, $\langle \vec{n}_s, \vec{u}_{g,s} \rangle$ are encoded using sixteen radial basis Gaussian functions, and used as scalar edge features in the bipartite graph. We follow a similar protocol for edges going from a vertex to a graph node.

## C.2 MODEL ARCHITECTURE DETAILS

In this section, we give more details on architectures that were explored during our investigations. All models rely on a PyTorch Geometric (Fey & Lenssen, 2019) implementation.

Our initial attempt involves a sequential alternation between surface and graph encoding, referred to as the `sequential` setting. This approach involves a two-step block: first, surface encoding is performed and its features are projected onto the graph via message passing, denoted as $h_n^g = \mathrm{MP}_{sg}(s_\theta(\mathcal{X}))_n$. Denoting the intermediate graph node embeddings as $\mathcal{H}^g = \{h_n^g, n \in \mathcal{V}_g\}$, they are propagated within the bipartite graph to derive surface embeddings again, using $\mathcal{X}_{out} = \alpha \mathcal{X} + \mathrm{MP}_{gs}(g_\theta(\mathcal{H}^g))$, where $\alpha \in \mathbb{R}$ is a residual connection. The architecture proposed by (Somnath et al., 2021) fits within this sequential framework, employing just one block and sum pooling for message passing.

However, the sequential approach does not simultaneously leverage the distinct scales (local and global) offered by graph and surface processing. To overcome this, the `bipartite` approach processes graph and surface features concurrently in separate encoders, and merge them with a message passing incorporating learnable parameters, based on the equation: $\mathcal{X}_{out} = \alpha \mathcal{X} + \mathrm{MP}_{gs}(\mathcal{H}) + \mathrm{MP}_{sg}(\mathcal{H})$. Those architectures are illustrated in Figure 4.

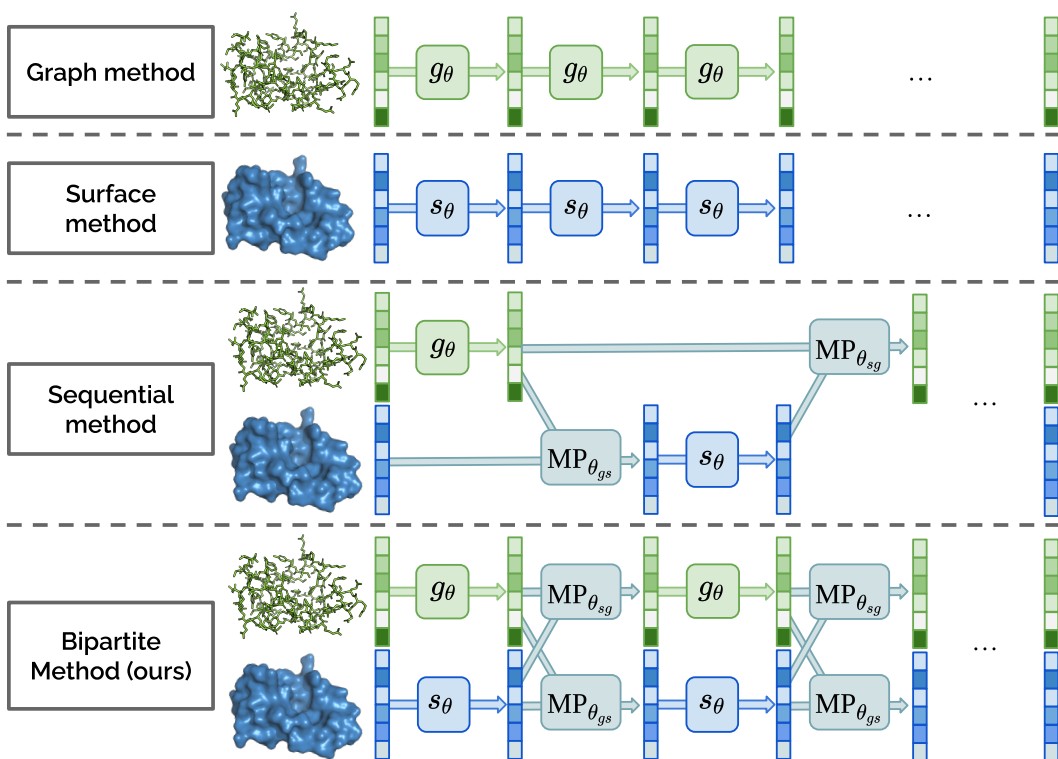

Supplementary Figure 4: Illustration of our approach integrating surface and graph information.

The architecture we used is outlined in the main text. We employ our modified version of DiffusionNet (by adapting the original implementation provided by the authors[1]) for each surface encoder and utilize GCN for the graph networks (using the implementation provided by PyTorch Geometric[2]). The surface methods were trained using only the surface encoder, whereas the mixed methods

---

[1]https://github.com/nmwsharp/diffusion-net
[2]https://pytorch-geometric.readthedocs.io/en/latest/

incorporated an additional graph encoder and a message-passing framework. When employing both encoders, we ensure the number of channels is equal for both. Thus, the variables are the number of "blocks" and the number of channels for each block. Our networks were trained in accordance with the parameter counts of other methods, strictly adhering to their optimization protocols, including the number of epochs, learning rate, and batch size.

For the surface methods, we consistently used 3 blocks, with 94, 90, and 96 channels for the PIP, MSP, and PSR tasks, respectively. For the bipartite methods, on the PIP task, we utilized 4 blocks with a width of 118; for MSP, 3 blocks with a width of 148; and for PSR, 4 blocks with a width of 160. On the binding site classification task, our architecture featured 6 blocks, each with a width of 128. The repository used to conduct experiments and reproduce these results is enclosed and will be publicly released upon acceptance.

# D  COMPUTATIONAL ANALYSIS

## D.1  COMPLEXITY ANALYSIS

In a DiffusionNet, the first main operations consist in a diffusion step using the equation introduced in the main text: $\mathbf{f}_t = \Phi e^{-\Lambda t}(\Phi^T M)\mathbf{f}_0$, for each feature map. For a given hidden dimension $h$, the complexity of this operation is hence $\mathcal{O}(h|\mathcal{V}_s|^\omega)$ with $\omega$ the matrix multiplication complexity. Then, the two subsequent operations are the pointwise gradient construction and mixing, as well as a point-wise MLP of complexity: $\mathcal{O}(h^2|\mathcal{V}_s|)$. Message passing layers complexity with regards to $h$ and their number of edges N is $\mathcal{O}(h^2 N)$. In our case, $N = 16|\mathcal{V}_s|$ for the bipartite graph.

Given additional time constants in a $k_{\text{diff}}^{(1)}, k_{\text{diff}}^{(2)}, k_g, k_{bp}$ representing the time needed for spectral and point-wise operations in *DiffusionNet*, graph encoding operations and message passing over the bipartite graph, the overall model complexity is $\mathcal{O}(k_{\text{diff}}^{(1)}h|\mathcal{V}_s|^\omega + (k_{\text{diff}}^{(2)} + 16 * k_{bp})h^2|\mathcal{V}_s| + k_g h^2|E|)$.

The spectral operation component $(k_{\text{diff}}^{(1)})$ does not depend strongly, but it has a strong dependency on the size of the mesh, which is not well addressed by reducing the number of parameters. Empirically, for the latent dimensions considered, this spectral component was the time bottleneck and thus, coarsening the graphs resulted in significant speedups. We adjusted the number of vertices to make the first term of similar magnitude to the others (obtained for a 0.1 coarsening rate). In this regimen, when dividing the parameter count equally between the three learnt components, we observed that the ProNet execution was the limiting operation, and hence conclude that our method has a similar throughput to graph-based method for a fixed parameter count. Moreover, we observed that our method seemed to require less parameters overall, and note that the parameter count is split between components, effectively resulting in smaller ProNets than in the pure ProNet network of the benchmark, and hence a slightly faster runtime overall.

## D.2  MEMORY AND BATCHING

From the storage and memory perspective however, the memory footprint of the surface-related operations dominate the ones originating from the graphs. We did not find it to be limiting in terms of I/O, but it effectively forbids the use of large batch sizes, which can be limiting.

In addition to individual objects being large, the large variance in the object size (see Figure 5) also prevents us from using large batch sizes. Indeed, due to the random composition of the batch, some unlucky batches contain several particularly large proteins, causing spikes in memory usage and out-of-memory errors. We validate in Figure 6 that using coarsened surfaces, we stay in the linear dependency on the number of vertices for the hidden sizes at stake.

This is illustrated in Figure 7, for which we tracked the memory consumption of our model through one epoch of learning. Observe the memory peaks that can result in over 20% excess memory consumption, keeping in mind that this is only one run over one epoch. When training several models for hours, the probability that one outlier results in a 50% increase forces user caution.

A naive solution is to use a safety margin, aggravating our small batch size issue. We implemented a dynamic batching procedure to create batches that are balanced in size, alleviating the memory

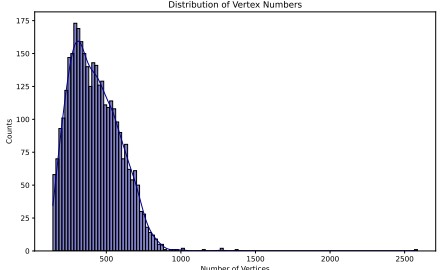

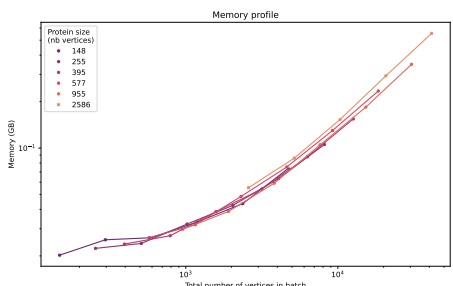

Supplementary Figure 5: Distribution of the number of vertices in a dataset.

Supplementary Figure 6: Memory consumption plotted as a function of the number of vertex, in a log-log scale.

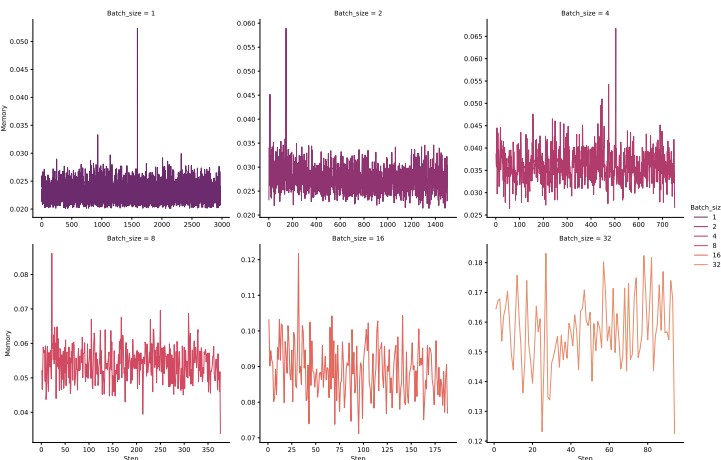

Supplementary Figure 7: Memory consumption monitoring through an epoch, for different batch sizes.

issues originating from batch size variability. Finding ways to reduce the memory footprint of surface networks remains an important direction.

### D.3 COMPUTING RESOURCES USED FOR THE PROJECT

Depending on the task, the training takes between a few hours and up to four days (for PIP which is our largest data set) on a standard setting; 4 CPU workers and a single GPU such as NVIDIA V100. To achieve our results, and our ablation experiment, we performed about 100 model training (5 data sets, several settings per datasets and some failed experiments).

## E VALIDATION DATASET DESCRIPTION

**RNA segmentation :**   This task aims to segment 5s ribosomal RNA molecules into functional components, as defined by aligned regions in a Multi-Sequence Alignment (Poulenard et al., 2019). This dataset consists of 640 RNA surface meshes of about 15k vertices, split at random. It was introduced as a benchmark to compare surface encoders.

**Protein Interaction Prediction (PIP) :**   This task aims to predict which part of a protein interacts with which part of another. Framed as a classification task, pairs of residues from two proteins are labeled as positives if they interact and as negatives if they do not. The dataset comprises 87k, 31k,

and 15k training, validation, and test examples, split based on a 30% sequence identity. Moreover, proteins in the test set are presented in their apo conformation.

**Mutation Stability Prediction (MSP) :** The objective here is to determine if a mutation enhances the stability of protein-protein interaction. Given a protein-protein interaction structure and its mutated version, this classification task labels the pair as a positive example if it exhibits increased stability. This task includes 2864, 937, and 347 examples in each data split, and the splitting is performed based on a 30% sequence identity. For both PIP and MSP, the performance metric is the Area under the Receiver Operating Characteristic curve (AuROC).

**Protein Structure Ranking (PSR) :** PSR is a regression task and aims to assign a quality score to predicted protein structures from the Critical Assessment of Methods of Protein Structure Prediction (CASP) (Kryshtafovych et al., 2019) competition. The PSR data train, validation, and test splits hold 25.4k, 2.8k, and 16k systems respectively. Splits correspond to a time split, with more recent CASP competition belonging to the test split. The "$R_g$" term represents the mean correlation across all systems and proposals. Meanwhile, "$R_l$" refers to the average correlation for each system.

**Masif-ligand**: This task aims to predict ligand-binding preferences for protein binding sites, as introduced in (Gainza et al., 2020). Binding sites are regions of protein surfaces where small molecules (ligands) interact with the protein. A fine characterization of those regions helps understanding protein functions and plays a significant role in drug design and discovery. This dataset consists of protein binding sites that accommodate one of seven co-factors. It comprises 1,634 training instances, 202 validation instances, and 418 test cases, split using sequence identity. Given a binding site, the task amounts to predicting its corresponding co-factor.

**AbAg**: This dataset focuses on binding site prediction in the context of immunology: predicting the interaction between an antibody and its target, denoted as the antigen. Improved understanding and ability to predict such interactions has direct applications in antibody-based treatments. (Pegoraro et al., 2024) introduces a dataset containing 235 antibody–antigen complexes, with 186 for training and 49 for testing, split based on a 70% similarity threshold. The task consists in predicting the residues involved in the binding site on the antibody and antigen separately. The performance on the antibody side is computed only on the Complementarity Determining Regions (CDRs). They propose different methods and we chose to compare against *MIXmean-egnn PiNet* that had the most balanced performance.

**PINDER**: This is a recent, large-scale dataset (Kovtun et al., 2024) that holds over 2M interaction in its full version. We used the clustered version of this dataset that holds around 42k structures and reduces redundancy. Pinder also introduces a strict splitting criterion based on interface similarity, that was shown to reduce leakage. In addition to their bound form (*holo*), systems in the test set are also available in their unbound form (*apo*) and as predicted by AlphaFold (*predicted*), representing a more realistic use case. The number of systems present in the test set is 1955 clusters, with 342 corresponding *apo* and 1747 *predicted* structures. This is two orders of magnitude higher than AbAg.

We introduce two tasks related to binding site prediction. Our first task is formulated as PIP (*Pinder-Pair*): given two interacting monomers taken in isolation, it aims at classifying pairs of residues into interacting, and non-interacting. The other task is close to Masif-Site (Gainza et al., 2020), only taking one protein as input to predict interacting residues (*Pinder-site*).

## F SUPPLEMENTARY RESULTS

### F.1 VALIDATION OF THE NEW MODEL ON PSR

We conduct a similar analysis of the proposed enhancements to the one on ribosomal RNA on the PSR task. We depict the obtained learning curves in Figure 8 and report performance in Table 1. Again, we observe that our suggested enhancements result in a more stable learning.

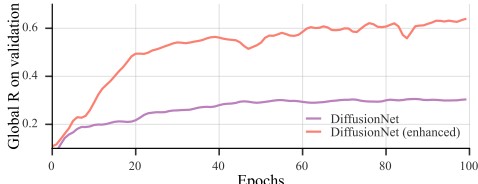

| | PSR | |
|---|---|---|
| | $R_l$ | $R_g$ |
| DiffusionNet (original) | 0.17 | 0.39 |
| DiffusionNet (enhanced) | **0.33** | **0.643** |

Supplementary Figure 8: Global correlation learning curve on the PSR task before and after applying our enhancements to *DiffusionNet*.

Supplementary Table 1: Results of our enhanced model on the PSR task.

## F.2 VARIATION OF OUR RESULTS OVER DIFFERENT SEEDS

In table 2, we computed the standard error of the mean computed over three replicates. The standard error is small compared to the gap of performance with the next best method, hinting for a statistically significant difference (unfortunately, there are no replicates for other methods to perform actual statistical tests). As expected, the standard error is higher for smaller datasets (MSP, AbAg) than larger ones (PIP, PSR).

Supplementary Table 2: Standard error of our model evaluated on three replicates for each of the task included. As a reference, we include the best performing competing model, without pretraining.

| Task | PIP | MSP | PSR | |
|---|---|---|---|---|
| Metric | AuROC | AuROC | $R_l$ | $R_g$ |
| *Runner-up method* | *87.1* | *68.5* | ***63.2*** | ***86.8*** |
| Ours - Mean | **90.9** | **71.5** | 61.7 | 85.7 |
| Ours - standard error | 0.088 | 1 | 0.27 | 0.22 |

| Task | Ab | | Ag | | Masif Ligand |
|---|---|---|---|---|---|
| Metric | AuROC | MCC | AuROC | MCC | Balanced acc. |
| *Runner-up method* | *80* | *28* | ***72*** | ***18*** | *81* |
| Ours - Mean | **88** | **53** | 67.3 | 14.3 | **88.2** |
| Ours - standard error | 1 | 2.1 | 1.8 | 1.5 | 0.3 |

## F.3 PERFORMANCE ON PINDER TASK

Table 3 shows the performance of `AtomSurf`, compared to ProNet on the clustered PINDER dataset. In this dataset, the systems are clustered to remove redundancy. Moreover, the splitting strategy is stricter, based on a similarity score of the interfaces. In addition to the canonical test set, the test set systems are also available in their unbound form (*apo* setting) and as predicted by AlphaFold (*predicted* setting). This results in a more realistic estimate of the quality of the predictions. We report the accuracy and AuROC on both tasks, on each of those splits. Finally, we also report its performance without ESM embeddings.

Supplementary Table 3: Performance on the Pinder Task.

| Task | *Pinder-Pair* | | | | | |
|---|---|---|---|---|---|---|
| Split | Holo | | Apo | | AF2 | |
| Metric | Auroc | Acc | Auroc | Acc | Auroc | Acc |
| Pronet | 80.1 | 72.5 | 78.2 | 71.5 | 73.5 | 67.9 |
| AtomSurf | 92.8 | 85.3 | **88.4** | **80.7** | **87.1** | **80.2** |
| AtomSurf - no ESM | **93** | **85.4** | 87.6 | 80.2 | 87 | 79.9 |

| Task | *Pinder-Site* | | | | | |
|---|---|---|---|---|---|---|
| Split | Holo | | Apo | | AF2 | |
| Metric | Auroc | Acc | Auroc | Acc | Auroc | Acc |
| Pronet | 74.3 | 69.2 | 70.7 | 71 | 60.6 | 56.3 |
| AtomSurf | **88.3** | **82.3** | **84.2** | **80.8** | **82** | 76.6 |
| AtomSurf - no ESM | 87.6 | 79.2 | 82.4 | 77.1 | 80.9 | **78.8** |

## F.4 MODEL ABLATION

Here, we present the detailed ablation study of our method `AtomSurf-bench`. We examine the impact of different design choices on our tasks. All of these results are obtained in benchmark settings, both in terms of features (only atom-type as input) and of surface encoders and graph encoders (GCNs).

First, we look at the overall architecture of our approach, following Appendix C, where we introduce the `sequential` and `bipartite` scenarios, which amount to variations in the connectivity of the blocks. In particular, the HoloProt method proposed in Somnath et al. (2021) falls into the *sequential* framework with just one encoding block and a MeshCNN (Hanocka et al., 2019) for protein encoder. For fairness, we replace MeshCNN with our encoder and refer to this setup as `HoloProt`.

Another major design choice is the choice of the Message Passing (MP) component. We explore the use of three possible message-passing networks. Motivated by the success of DGCNN (Wang et al., 2019), in our `Att.` setting, we discard the geometric notion of a neighborhood, allowing for potentially long-distance message passing. In this setting, all nodes from the graphs attend to all vertices from the surface. To deal with the incurred computational burden, we use the recent memory-efficient Flash Attention (Dao et al., 2022). We also explore the use of more conventional Graph Convolutional Networks (`GCN` setting) (Kipf & Welling, 2016) and the use of Graph ATtention networks (Veličković et al., 2017; Brody et al., 2021) for our final `GAT` setting. Finally, we also try using three or four blocks in our networks, always adjusting the network width to keep the number of parameters constant. Our results are presented in Table 4.

Supplementary Table 4: Ablation study of our method: We compare various architectural designs, and message-passing methods for our task.

| Method | MP | MSP AuROC | PIP AuROC | PSR local R | PSR global R |
|---|---|---|---|---|---|
| sequential | - | 0.609 | 0.855 | 0.319 | 0.71 |
| HoloProt | - | 0.537 | 0.824 | 0.383 | 0.715 |
| | Att. | 0.689 | 0.791 | 0.402 | 0.792 |
| bipartite | GCN | 0.697 | 0.868 | 0.421 | 0.797 |
| | GAT | **0.707** | **0.876** | **0.452** | **0.833** |

The `sequential` strategy displays underwhelming results, which could root from its incapacity to handle multi-scale, simultaneous message passing. Similarly, `HoloProt` does not display a top performance, suggesting that their results could be enhanced by using better-performing mixing strategies.

Among the `bipartite` settings, `Att.` is consistently outperformed by the localized message-passing networks. The other scenarios give an overall close performance, with an edge for the `GAT` network. This is especially true on PSR, where surface methods alone were failing. The mixed approach could simply use surface diffusion as an efficient long-distance communication, which could explain why an attentive mechanism results in a performance boost in this scenario.

In short, the best setting leverages several blocks of message passing, with enhanced results for the most interconnected networks. In addition, the geometry of the bipartite graph is important, and an attentive mechanism generally benefits the optimal mixing.

## F.5 ESM EMBEDDINGS ABLATION

We conducted an Ablation of our method by training models without the ESM embeddings as input features. We do not change hyperparameters of our method to make up for this change, which strongly disrupts the training on smaller datasets, and especially on MSP which fails to train. We report the result in Table 5 on all tasks but Pinder, and in Table 3 for the PINDER task.

Supplementary Table 5: Performance of our model when ablating the ESM node embeddings.

| Task | PIP | MSP | PSR | |
|---|---|---|---|---|
| Metric | AuROC | AuROC | $R_l$ | $R_g$ |
| Runner-up method | 87.1 | 68.5 | 63.2 | **86.8** |
| Ours - Mean | **90.9** | **71.5** | **61.7** | 85.7 |
| Ours - no ESM | 90 | 54.5 | 47 | 82 |

| Task | Ab | | Ag | | Masif Ligand |
|---|---|---|---|---|---|
| Metric | AuROC | MCC | AuROC | MCC | Balanced acc. |
| Runner-up method | 80 | 28 | **72** | **18** | 81 |
| Ours - Mean | **88** | **53** | 67.3 | 14.3 | **88.2** |
| Ours - no ESM | 86 | 48 | 68 | 14.5 | 84 |

The performance decreases as expected, showing that ESM is truly an important feature. However, even without ESM, our method still shows significant results on most tasks, such as PIP, AbAg and MasifLigand. The overall difference was more noticeable on smaller datasets, especially on MSP for which the ablated network failed to learn a useful signal without ESM. Considering that AtomSurf-bench (which does not use ESM embeddings) gets 70.9 AuROC on MSP, we believe a high performance could be recovered by tuning the optimization. Based on this ablation, we conclude that ESM features are an important source of information, but that even without them, integrating a surface and a graph method gives a strong performance.

On PINDER, we obtain a small benefit from using ESM embeddings as node features, most notably on the apo set. However, this gap seems to be quite small on this dataset with stricter splits. One possible explanation for this result would be that those sequence-derived features contribute to memorization, whilst learnt structural properties could better generalize.

## F.6 ASSESSMENT OF THE IMPACT OF THE THRESHOLD DISTANCE

To evaluate the sensitivity of our results to variations of the distance threshold used to construct the bipartite graph, we varied this parameter on the PSR task. We report our findings in table 6. Our investigation indicates that changes to the threshold have a moderate effect on the model's performance. However, it is important to note that excessively large values may lead to out-of-memory errors.

Supplementary Table 6: Performance on the PSR task as a function of the threshold used.

| Threshold (A) | 4 | 8 | 10 | 20 |
|---|---|---|---|---|
| local R | **0.445** | 0.434 | 0.434 | OOM |
| global R | 0.812 | **0.833** | 0.815 | OOM |

