# OpenReview forum: "AtomSurf: Surface Representation for Learning on Protein Structures"
_ICLR.cc/2025/Conference — ICLR 2025 Poster_

### Official Review · Reviewer_dxE8 · 2024-10-29

**Soundness:** 4
**Presentation:** 4
**Contribution:** 2
**Rating:** 6
**Confidence:** 5

**Summary:**

The paper proposes a new model for protein structure representation learning and property prediction. The main idea is to take as input both the 3D point cloud graph as well as surface mesh, jointly process both modalities within the network with information sharing among them, and then use the joint representation to perform prediction tasks per residue or globally. Experimental results show improvements or match state of the art results on all benchmarks evaluated.

**Strengths:**

- The general ideas of share information across the various representations of a biomolecule is simple and appealing. Implementing a tighter coupling between the graph and surface track in the newly proposed networks is a sensible idea. The results support all these design decisions well. I think the community will find the ideas and results in the paper to be useful.

- Well written and well presented overall.

**Weaknesses:**

- There has been a recent focus in the community on purely sequence-based approaches and protein language models (see the last ICLR for several works on structure-aware pLMs). I would have been interested to see empirical comparisons to at least ESM2 or perhaps more recent structure-aware pLMs. In principle, the sequence contains all the information that represents a protein and pLMs could implicitly be able to learn surface representations. I would at least have liked to see an ablation on the use of ESM2 embeddings in the model.
    - (In other words: Could the performance gain be coming from the use of ESM2 embeddings as node features? Perhaps yes, as these pLM embeddings have been shown to already be very strong at learning structural representations and for structure-based tasks: https://openreview.net/forum?id=sTYuRVrdK3, ICLR 2023.)

**Questions:**

- Are the tables missing more recent results on ATOM3D? Eg. Table 1 only shows the results from the proposed models as well as the original ATOM3D benchmark. But there have been many other papers which have outperformed the original benchmark results. Is the rationale for not including them that they do not strictly conform to the benchmark protocol?

- I realise that the results on all the benchmarks show improvements and each table has the new model in bold -- this is good -- but how relevant are some of the ATOM3D and other tasks in the current state of the field? Eg. Tasks like PSR seem irrelevant in the post AlphaFold 2 era. For RNA segmentation: I personally don't think this is relevant for real-world problems, but I am  happy to be shown wrong here.

---

Minor comments:
- Line 287: typo 'embed'

---

> ### Author Response · Authors · 2024-11-21
> **Answer to reviewer 4**
>
> Thank you for your comments. We would like to raise to your attention that we also added a general answer in which we detail additional results obtained for the rebuttal. Shortly put, we added seed replicates to our results, an ablation study of the protein language features and a new result on the PINDER dataset. In addition, we provide a point by point answer to your comments below.
>
> >There has been a recent focus in the community on purely sequence-based approaches and protein language models (see the last ICLR for several works on structure-aware pLMs). I would have been interested to see empirical comparisons to at least ESM2 or perhaps more recent structure-aware pLMs. In principle, the sequence contains all the information that represents a protein and pLMs could implicitly be able to learn surface representations. I would at least have liked to see an ablation on the use of ESM2 embeddings in the model. (In other words: Could the performance gain be coming from the use of ESM2 embeddings as node features? Perhaps yes, as these pLM embeddings have been shown to already be very strong at learning structural representations and for structure-based tasks: https://openreview.net/forum?id=sTYuRVrdK3, ICLR 2023.)
>
> Thank you for the suggestion. We included this ablation, and overall show that removing those embeddings decreases, but retains most of the performance of our approach. Hence, those features are helpful but the protein structure, and proposed method still significantly contributes to the performance of our approach.
>
> >Are the tables missing more recent results on ATOM3D? Eg. Table 1 only shows the results from the proposed models as well as the original ATOM3D benchmark. But there have been many other papers which have outperformed the original benchmark results. Is the rationale for not including them that they do not strictly conform to the benchmark protocol?
>
> In Table 1, the rationale is indeed to only compare representations in the fairest way possible, which is why we compared those methods to Surface representation and to Atomsurf-bench. This was the spirit in which the ATOM3D paper came to be. This table gave us the insight that surfaces in isolation were not top-performers, but that the surface/graph combination was worth investigating
> However, in Table 2, we include more recent benchmarks that compare holistic encoders beyond their initial representations.
>
> > I realise that the results on all the benchmarks show improvements and each table has the new model in bold -- this is good -- but how relevant are some of the ATOM3D and other tasks in the current state of the field? Eg. Tasks like PSR seem irrelevant in the post AlphaFold 2 era. For RNA segmentation: I personally don't think this is relevant for real-world problems, but I am happy to be shown wrong here.
>
> We agree with the reviewer that the RNA segmentation task has little biological impact, even though domain segmentation might be useful for tasks like structural homolog mining. However, this task was previously proposed in the surface community, and the shapes it represents have more similarity to protein surfaces than, say, chairs and airplanes (that are the standard in 3D computer vision encoding). Therefore, we found this dataset to be relevant to compare several surface-based encoders.
>
> Protein folding is not yet a fully solved problem, so we believe model quality assessment is still relevant nowadays. We agree that a more modern instance of the PSR task would give us better insights with regards to our model’s ability to score AF3 folded proteins and hence provide alternatives to pLDDT scores.
>
> Our validation also includes assessing the impact of mutation on function, as well as interaction-related tasks, which are of relevance for biology.
>
> In addition to these tasks, we also included an assessment of our performance on the PINDER dataset, which is a more up-to-date task, and details our performance in our general comment.
>
> Overall, this comment is close to the previous reviewer’s comment about benchmark tasks in general. While we acknowledge that those tasks are not perfect, they are useful proxies to compare protein structure encoders in a fair way. The best validation will stem from the use of this encoder in actual applications, on which we are already working. We added a sentence in the limitation and perspectives paragraph
>
> &nbsp;
>
> We have tried to address all of the comments and questions by the reviewer, and thank the reviewer again for helping us improve our submission. We invite the reviewer to also consider the general feedback we provided in our general comment to all reviewers. As mentioned in our general comment, we will make sure to address *all* comments or concerns in the potential final version of our paper.

---

> ### Comment · Reviewer_dxE8 · 2024-11-25
>
> Thanks for the response. I retain my assessment and score, which I think reflects my current view of the paper's contributions and experimental setup.
> - Soundness: 4
> - Presentation: 4
> - Contribution: 2
> - Overall: 6/10
>
> Thanks for showing some results ablating the use of ESM embeddings. My takeaway is a bit different from yours: I thought that they do seem important for the model's performance and that the overall results without ESM embeddings are not outright improvements on the state of the art. I don't think one always needs to have SOTA results to have a great contribution, but I think the idea of combining a graph and surface representation is not super novel -- so it needs outright better modelling to convince me in its current state. I think the paper would improve with contextualization w.r.t. structure-aware pLMs like SaProt (ICLR'24 spotlight) and ProsT5, either via experiments or at least mentioning that the community is super interested in this new direction. I am still voting to accept the paper.
>
> Note: Authors have **not** updated the PDF of their submission despite promising to include many new results and new datasets. OpenReview comments may not provide the necessary details about experimental setup. I realise its my mistake for commenting so late in the rebuttal period (hectic period), but I think authors should update the PDF with new experiments and setup details at least for the Area Chair's decision.

---

> ### Author Response · Authors · 2024-11-25
> **Thanks for your answer**
>
> We thank you for your answer and your positive vote, and appreciate your comments.
>
> Please note that the ablated version of our method does retain state-of-the-art performance on most tasks, notably on interaction tasks on PIP (90 vs 87.1), Masif-Ligand (84 vs 81), AbAg task (48 vs 28 on the MCC Ab metric, similar on others) and Pinder (overall ~+10 AuROC points). We will make sure to provide a better contextualization in the final version.
>
> We agree that the use of structure-informed pLM (like SaProt or Prost5) is a very interesting direction. We will add a reference to these approaches in our introduction.
>
> We are in the process of updating our manuscript, our current struggle is the page limit, but we will post an updated pdf before the end of the rebuttal period.
>
> We thank the reviewer again for his or her input.

---

> ### Comment · Reviewer_dxE8 · 2024-11-25
>
> While other reviewers seem convinced by the rebuttal, I think seeing the revised PDF is pretty important to me at this stage (and is something that is generally expected at ICLR) given that there are significant new experiments.

---

> ### Author Response · Authors · 2024-11-27
> **Submission updated !**
>
> We have uploaded an updated pdf version of our submission. This update compiles the reviewers' feedback and we believe it made our paper stronger, thanks for that !
>
> We hope that we answered your concerns and would be happy to answer any additional questions you might have.

---

> > ### Comment · Reviewer_dxE8 · 2024-11-28
> >
> > Thanks for updating the PDF with new experiments.
> >
> > I believe this paper should be accepted and is heading that way. I stand by my score of 6 but have increased my confidence to 5.
> >
> > The paper is technically sound and well presented. My score is not higher because (1) I don't think a lot of the benchmarks used are biologically relevant; and (2) the methodological contribution of combining a graph and surface representation within middle layers of a network is not sufficiently novel in my opinion.

---

### Official Review · Reviewer_LVjj · 2024-11-03

**Soundness:** 3
**Presentation:** 4
**Contribution:** 3
**Rating:** 8
**Confidence:** 4

**Summary:**

## Analysis of surface representations methods in protein representation learning

This paper studies the utility of a diffusion kernel based surface representation ideas, introduced in DiffusionNet (Sharp et al. 2020 preprint, 2022 published) for protein representation learning. The authors update the original DiffusionNet formulation to (1) allow for batching and (2) initialising the diffusion time parameters around a meaningful value in proteins (10 Angstrom).

This protein-adapted version of DiffusionNet is then studied (1) in isolation and (2) in combination with residue graph-based representation learning on a handful of protein representation learning tasks:

Evaluated are the protein interaction prediction (PIP), mutational stability prediction (MSP) and protein structure ranking (PSR) tasks of the Atom3D benchmark, as well as an antibody-antigen interaction prediction and a protein-ligand binding site prediction task.

The contributions of this work are:
- (1) *Modelling contributions*: The above mentioned (batching, initialisation) adaptations to DiffusionNet and, for the models that work on both surface and residue graph level representations, a layer-wise cross-communication between surface & graph representations via cross attention.
- (2) *Insight contributions (ablations and benchmark comparisons)*: The authors ablate the use of DiffusionNet-style surface representation learning versus other surface based approaches, as well as combining surface and graph level approaches. They compare to adequate baseline representatives of each approach that are common in the literature. They find that in their hands & on the above mentioned benchmarks their DiffusionNet-style approach together with a residue graph representation based approach outperforms the other methods.

**Strengths:**

1. Writing: The paper was very well written overall and I enjoyed the read. Statements were clearly made and easy to follow -- well done on the writing and thank you for spending time on it!

2. Benchmarks select relevant baselines: The authors selected relevant baselines (e.g. GearNet for graph-level representation learning, HoloProt for graph-surface combinations, MaSIF for surface based representation learning) that I would have expected to see in such a work.

3. Interesting case of cross-domain method transfer: The authors transfer the DiffusionNet method from the general surface representation learning literature to the protein representation learning literature. As someone from the protein community I was not previously familiar with the DiffusionNet work and found this a valuable bridge and interesting comparison. While the adaptations to the DiffusionNet approach for proteins were minor (essentially constrained to the update of sampling the t values at 10 Angstrom) this adaptation was well reasoned and makes sense from a domain-level perspective.

4. Ablations: I found the ablations to be informative and well constructed and described.

**Weaknesses:**

1. Clarity in benchmark comparisons: When seeing the comparison to the benchmark methods, I was left with the question whether the authors re-ran training on the same datasets & splits for each method, or whether these numbers were quoted from a table? What are the levels of variation for each? (e.g. +/- resulting from 3-5 different runs). Were all approaches 'tuned' with equal attention to hyperparameters? I would have expected a short discussion of these points.

2. Limitation of these benchmarks: This is less of a direct critique of this paper, and more of the underlying benchmarks that were used (that I believe should be updated). Protein structures tend to not be rigid, and as a consequence surfaces tend not to be rigid either. While at a coarse grained level there is likely less variation (unless there are global conformation shifts), at a 3-8A level there can be significant reorganisation of side chains - relevant for ligand binding and polar interactions. Interaction prediction should therefore be done on 'relaxed' surfaces that were not extracted from protein complexes (as some of the benchmarks do) and holdout sets should be constructed in particular also based on structural similarities of surfaces rather than just sequence homology. I would highly recommend the authors to look into the pinder & plinder datasets & benchmarks, but I do acknowledge that these were just published briefly before this work was submitted. A comparison on these in the rebuttals would be of much interest. Without comparisons like these I doubt the real-world applicability of such surface representations for most practical tasks in biology.

**Questions:**

C.f. weaknesses above.

---

> ### Author Response · Authors · 2024-11-21
> **Answer to reviewer 3 (part 1/2)**
>
> >Writing: The paper was very well written overall and I enjoyed the read. Statements were clearly made and easy to follow -- well done on the writing and thank you for spending time on it!
> Benchmarks select relevant baselines: The authors selected relevant baselines (e.g. GearNet for graph-level representation learning, HoloProt for graph-surface combinations, MaSIF for surface based representation learning) that I would have expected to see in such a work.
> Interesting case of cross-domain method transfer: The authors transfer the DiffusionNet method from the general surface representation learning literature to the protein representation learning literature. As someone from the protein community I was not previously familiar with the DiffusionNet work and found this a valuable bridge and interesting comparison. While the adaptations to the DiffusionNet approach for proteins were minor (essentially constrained to the update of sampling the t values at 10 Angstrom) this adaptation was well reasoned and makes sense from a domain-level perspective.
> Ablations: I found the ablations to be informative and well constructed and described.
>
> Thank you for your very positive feedback and your comments. We would like to raise to your attention that we also added a general answer in which we detail additional results obtained for the rebuttal. Shortly put, we added seed replicates to our results, an ablation study of the protein language features and a new result on the PINDER dataset. In addition, we provide a point by point answer to your comments below.
>
> >Clarity in benchmark comparisons: When seeing the comparison to the benchmark methods, I was left with the question whether the authors re-ran training on the same datasets & splits for each method, or whether these numbers were quoted from a table? What are the levels of variation for each? (e.g. +/- resulting from 3-5 different runs). Were all approaches 'tuned' with equal attention to hyperparameters? I would have expected a short discussion of these points.
>
> We completely agree that this is a potential high source of bias, often found in scientific papers. Indeed, re-implementing competitors’ methods with little care for tuning is the easiest way to shine. For fairness, we thus only provide results that were externally reported by original papers or independent implementations. We emphasize that we also stuck to their learning settings (with the exception of learning rate adjustments) and kept the original problem formulations, only changing the encoders. For instance, on the antibody-task, the cross talk between proteins was taken as is from the original paper. This indicates that our approach is flexible enough to accommodate those different tasks and that our performance boosts are truly attributable to enhanced protein structure encoding.
> We added a clarification in the text.
>
> We also added an assessment of the robustness of our method by training several replicates and added this result in the text. We also added this Table in the general comment (point 1).

---

> ### Author Response · Authors · 2024-11-21
> **Answer to reviewer 3 (part 2/2)**
>
> >Limitation of these benchmarks: This is less of a direct critique of this paper, and more of the underlying benchmarks that were used (that I believe should be updated). Protein structures tend to not be rigid, and as a consequence surfaces tend not to be rigid either. While at a coarse grained level there is likely less variation (unless there are global conformation shifts), at a 3-8A level there can be significant reorganisation of side chains - relevant for ligand binding and polar interactions. Interaction prediction should therefore be done on 'relaxed' surfaces that were not extracted from protein complexes (as some of the benchmarks do) and holdout sets should be constructed in particular also based on structural similarities of surfaces rather than just sequence homology. I would highly recommend the authors to look into the pinder & plinder datasets & benchmarks, but I do acknowledge that these were just published briefly before this work was submitted. A comparison on these in the rebuttals would be of much interest. Without comparisons like these I doubt the real-world applicability of such surface representations for most practical tasks in biology.
>
> We completely agree that the ability to detect binding preferences should be evaluated on apo conformations. The performance of structure-based tools should also assess on predicted structures (even though those predicted structures might be biased towards holo conformations). As mentioned though, there is a lack of benchmarks for those two aspects. We would like to emphasize that the PIP data set already performs its tests on apo forms of the complex. However, there are no structural comparisons of the interfaces involved in the data splitting.
>
> Therefore, we were enthusiastic about the release of the datasets you cited. We started processing those huge datasets before rebuttal and are therefore able to report a performance on the Pinder dataset. We basically propose two binding site detection tasks and report a performance well above ProNet. This performance is stable across holo, apo and predicted splits; and not very sensitive to ESM ablation. We provide more details and a table in the general answer (point 3).
>
> &nbsp;
>
> We have tried to address all of the comments and questions by the reviewer, and thank the reviewer again for helping us improve our submission. We invite the reviewer to also consider the general feedback we provided in our general comment to all reviewers. Given the positive appreciation of our work, we respectfully invite the reviewer to consider increasing their overall rating. As mentioned in our general comment, we will make sure to address *all* comments or concerns in the potential final version of our paper.

---

> > ### Comment · Reviewer_LVjj · 2024-11-25
> > **Review acknowledged**
> >
> > Dear authors,
> >
> > Thank you for your work in addressing and clarifying my points. Thank you in particular for adding results on the pinder dataset, which I am sure will be helpful for the field in the future as well as I anticipate this becoming a more standard benchmark.
> >
> > My main concerns have been addressed, and while I am still critical of the overall apo <> holo discrepancies in the field, I'm raising my score in acknowledgement of the author's answers to my review questions.
> >
> > I believe this is a helpful contribution to the field and should be accepted at ICLR2025.

---

> ### Author Response · Authors · 2024-11-25
> **Thank you !**
>
> Thank you for your answer and for raising your evaluation of our work !

---

### Official Review · Reviewer_f7eD · 2024-11-04

**Soundness:** 3
**Presentation:** 3
**Contribution:** 2
**Rating:** 5
**Confidence:** 3

**Summary:**

This work proposes a hybrid approach combining surface and graph-based encoders, an improved DiffusionNet architecture tailored for protein surface analysis, a comparison of protein surface-based representations against protein as graph-based representations and enhanced computational efficiency with residue graphs and coarsened meshes.

**Strengths:**

1. The proposed method shows superior empirical performance on Protein Interaction Prediction (PIP) and Mutation Stability Prediction (MSP) tasks compared to existing methods.
2. The paper is easy to follow.
3. The code is provided.

**Weaknesses:**

I believe the main weakness of the paper is that, in terms of the proposed method, the paper does not provide a significant novel contribution. While the authors argue that direct comparisons between protein surface representations and other forms of representation have not yet been conducted, the idea of using protein surface representations is not new. Additionally, the results for the Protein Structure
Ranking (PSR) task in the Atom3D benchmark demonstrate comparable performance with other approaches that incorporate different representations, such as GearNet and ProNet.

**Questions:**

1. The authors modified the design of the DiffusionNet architecture, which is typically used with geometric representations like triangle meshes or point clouds, and applied it at each point (or vertex). Initially, the protein surface was computed using MSMS, which generates a surface by rolling a virtual sphere (representing a solvent molecule, usually water) over the protein's atoms. Were traditional geometric CV surface reconstruction methods, such as Poisson surface reconstruction, considered to enhance compatibility with the adapted approach (DiffusionNet)?
2. "We choose to use residue graphs for computational efficiency: each node in this graph represents an amino-acid residue and edges are introduced between pairs of atoms that are within a 12 $\unicode{x212B}$ radius cutoff. For each residue, we add a one hot encoding of its type, its secondary structure annotation and its hydrophobicity. We also compute sequence embeddings using the ESM-650M (Rives et al., 2021) pretrained model and include them as node features." How were ESM embeddings used for amino acid residue atoms?
3. The surface encoder is based on the DiffusionNet architecture, which is applied to vertices (either from a point cloud or mesh). However, a more intuitive approach for capturing surface representation could be to use a triangulated mesh-based encoder defined on edges of the surface, such as MeshCNN. While DiffusionNet outperformed MeshCNN in the original study, it would be interesting to include an analysis of MeshCNN's performance within the protein domain.
4. The ablation study lacks a comparison between random representations and ESM residue representations.
5. Table 2 presents the number of parameters and indicates if pretraining was performed. If Atomsurf uses ESM embeddings, why it was not indicated as pretrained?

---

> ### Author Response · Authors · 2024-11-21
> **Answer to reviewer 2 (part 1/2)**
>
> Thank you for your comments. We would like to raise to your attention that we also added a general answer in which we detail additional results obtained for the rebuttal. Shortly put, we added seed replicates to our results, an ablation study of the protein language features and a new result on the PINDER dataset. In addition, we provide a point by point answer to your comments below.
>
> >The proposed method shows superior empirical performance on Protein Interaction Prediction (PIP) and Mutation Stability Prediction (MSP) tasks compared to existing methods.
>
> >I believe the main weakness of the paper is that, in terms of the proposed method, the paper does not provide a significant novel contribution. While the authors argue that direct comparisons between protein surface representations and other forms of representation have not yet been conducted, the idea of using protein surface representations is not new. Additionally, the results for the Protein Structure Ranking (PSR) task in the Atom3D benchmark demonstrate comparable performance with other approaches that incorporate different representations, such as GearNet and ProNet.
>
> Please note that we do not claim to present the first method that uses the surface representation. Instead, we 1. Evaluate for the first time the surface representation to other SOTA approaches within a well-established benchmark, and 2. Present a novel approach that combines surface-based and graph-based learning in a tightly integrated framework, which surpasses previous approaches. Both our rigorous evaluation of the most recent surface-based learning as well as our novel dual architecture are done for the first time, and thus represent the core contribution of our work.
>
> Regarding empirical performance, we would like to invite the reviewer to consider our reported results on antibody binding site prediction and on binding site classification
>
> >The authors modified the design of the DiffusionNet architecture, which is typically used with geometric representations like triangle meshes or point clouds, and applied it at each point (or vertex). Initially, the protein surface was computed using MSMS, which generates a surface by rolling a virtual sphere (representing a solvent molecule, usually water) over the protein's atoms. Were traditional geometric CV surface reconstruction methods, such as Poisson surface reconstruction, considered to enhance compatibility with the adapted approach (DiffusionNet)?
>
> To avoid any confusion, we note that we did apply DiffusionNet on a triangle mesh-based surface representation, which was produced by MSMS, as is commonly done in other works. Our adaptation of DiffusionNet within our integrated graph and surface-based framework enables information sharing between surface and graph-based representations across all (including intermediate) layers of the network and we use message passing between graph node and surface vertices to achieve this.
> While other surface reconstruction approaches, such as Poisson-based methods can lead to more accurate results and thus potentially more robust learning, we found the mesh quality produced by MSMS to be sufficiently good for our purposes. We attribute this in part to the robustness of DiffusionNet to mesh connectivity. We do agree that a proper evaluation represents an interesting direction for future work.
>
> >We choose to use residue graphs for computational efficiency: each node in this graph represents an amino-acid residue and edges are introduced between pairs of atoms that are within a 12 Å radius cutoff. For each residue, we add a one hot encoding of its type, its secondary structure annotation and its hydrophobicity. We also compute sequence embeddings using the ESM-650M (Rives et al., 2021) pretrained model and include them as node features." How were ESM embeddings used for amino acid residue atoms?
>
> For residue-level graphs, where nodes are residues, we simply add them as node features. For atom-level graphs, where nodes are atoms, we have two methods to incorporate residue-level features. Either we do not include them, or we assign those as constant features for all atoms in the same residue.
> In our paper, atom-level graphs are only used for Atomsurf-bench (that does not include ESM embeddings).

---

> ### Author Response · Authors · 2024-11-21
> **Answer to reviewer 2 (part 2/2)**
>
> >The surface encoder is based on the DiffusionNet architecture, which is applied to vertices (either from a point cloud or mesh). However, a more intuitive approach for capturing surface representation could be to use a triangulated mesh-based encoder defined on edges of the surface, such as MeshCNN. While DiffusionNet outperformed MeshCNN in the original study, it would be interesting to include an analysis of MeshCNN's performance within the protein domain.
>
> While we acknowledge how intuitive MeshCNN is, the implementation is actually slow and not tractable for large datasets like PIP and PSR, making this comparison hard to conduct in practice. Moreover, MeshCNN displays several possible limitations, the main of which is its sensitivity to the mesh connectivity, which echoes with your first question.
>
> >The ablation study lacks a comparison between random representations and ESM residue representations.
>
> We added an ESM ablation to our study. Shortly, we found that using ESM improves our results, but that AtomSurf still retains a top performance without those embeddings. Those results are detailed in the general answer.
>
> >Table 2 presents the number of parameters and indicates if pretraining was performed. If Atomsurf uses ESM embeddings, why it was not indicated as pretrained?
>
> Thank you for pointing this out, this point was unclear.
> We make a distinction between using features derived from a protein language model and using network weights pretrained on a structural task. The weights of our network are randomly initialized on each task. This is in contrast with other methods, like GVP-ESM that includes structure-based pretraining, for instance with denoising or contrastive learning.
> We corrected our table to make this point clearer.
>
> &nbsp;
>
> We have tried to address all of the comments and questions by the reviewer, and thank the reviewer again for helping us improve our submission. We invite the reviewer to also consider the general feedback we provided in our general comment to all reviewers. As mentioned in our general comment, we will make sure to address *all* comments or concerns in the potential final version of our paper.

---

> ### Author Response · Authors · 2024-11-27
> **Submission updated !**
>
> We have uploaded an updated pdf version of our submission. This update compiles the reviewers' feedback and we believe it made our paper stronger, thanks for that !
>
> We hope that we answered your initial concerns and would be happy to answer any additional questions you might have.

---

### Official Review · Reviewer_3ZDU · 2024-11-04

**Soundness:** 3
**Presentation:** 2
**Contribution:** 3
**Rating:** 8
**Confidence:** 5

**Summary:**

The paper aims to improve our understanding of surface-based learning for protein representations. This is achieved by first adapting the surface encoder from the graphics literature for protein-learning tasks, which is then compared against other representation modalities (graphs, convolutions, equivariance etc). Following this the authors consider learning on bipartite graphs comprising structure and surface, allowing information to be shared at every layer of learning, compared to previous multi-scale approaches.

Experiments on the Atom3D benchmark showcase the improved performance of the method on PPI (Protein-Protein Interaction), MSP (Mutation-Stability Prediction), and PSR (Protein-Structure Ranking) tasks, while using a much smaller model. A further experiment on binding site classification sees the method performance favorably compared to previous surface-learning based approaches.

**Strengths:**

1. There is very limited work on understanding representation learning with protein surfaces, even though it is universally agreed upon as being a suitable choice for many tasks. In that light, the paper is original in its attempt to dissect and improve representation learning with protein surfaces, and their integration with other modalities.

2. The quality of this work is good, showing a careful study of protein surfaces, adapting existing encoders to capture biological mechanisms, and experimental performance with relevant baselines on a well-known (albeit old) benchmark. The text is also clearly written, and generally easy to follow. For experiments, both the experimental comparisons against other representation modalities, and the experiments on antibodies are interesting and promising.

3. Ablation studies are also performed for the different experimental choices, which further helps improve our understanding of protein-surface learning.

**Weaknesses:**

**Experiments**

1. Atom3D is an old benchmark, without known issues with regards to the strategies used for splitting datasets. There has been some recent efforts [1] in curating tasks and datasets and evaluating structure based protein learning methods. While I understand carrying out an evaluation with this is not feasible within the scope of the current rebuttal timeline, it would be very handy to have this in a future revision of the paper.

[1] Evaluating Representation Learning on the Protein Structure Universe

2. For the BindingSite Identification experiment, there is a missing baseline from dMaSIF which also achieves a 0.87 AUROC.

3. For most of these tasks, the dataset sizes available are typically limited. This makes it paramount to include uncertainty estimates, measured across 3 experimental runs.

**Presentation**
The experiments section should include the corresponding table for the ribosomes (rather than in the appendix), or reorder the sections such that it falls under Ablation studies. This would improve the readability and flow of the paper.

**Questions:**

1. Could the author(s) include the uncertainty estimates, measured across 3 experiment runs. I presume running these would not be out-of-scope for the rebuttal timeline, given the small size of the model and the datasets.

2. For the different benchmark tasks, it would be very beneficial to understand / detail how the training splits were constructed. A common practice is to construct splits based on sequence similarity, but these don't capture the structural nuances / differences which are more relevant for OOD generalization.

---

> ### Author Response · Authors · 2024-11-21
> **Answer to reviewer 1**
>
> Thank you for your comments. We would like to raise to your attention that we also added a general answer in which we detail additional results obtained for the rebuttal. Shortly put, we added seed replicates to our results, an ablation study of the protein language features and a new result on the PINDER dataset. In addition, we provide a point by point answer to your comments below.
>
> >Experiments
> Atom3D is an old benchmark, without known issues with regards to the strategies used for splitting datasets. There has been some recent efforts [1] in curating tasks and datasets and evaluating structure based protein learning methods. While I understand carrying out an evaluation with this is not feasible within the scope of the current rebuttal timeline, it would be very handy to have this in a future revision of the paper.
> [1] Evaluating Representation Learning on the Protein Structure Universe
>
> While we agree Atom3d is not as recent as ProteinWorkshop, it has had some adoption, which allowed us to compare to official, externally reported performances for competitors. We think this is fairer than retraining models since it is easy to break other methods even with relatively minor changes, such as by using the wrong learning rate.
> We would like to comment on the concerns raised about the splitting strategies. In Atom3D, the splits rely on sequence identity for MSP and PIP (with cutoff 30%), which is also the case for several tasks in ProteinWorkshop (though with less stringent cutoff 50%), and on temporal splitting for PSR which is realistic.
>
> However, adding more tasks to our validation represents a valid, and interesting direction. The Metal Binding Site prediction and PTM tasks from Protein Workshop in particular caught our attention, and we will work on adding those to our validation. Among the tasks considered in recent benchmarks, we also found the PINDER dataset particularly rich and closely aligned with our evaluation framework. We have added validation on this recently introduced dataset, introducing a binding site task. Even though there is no reported performance from other methods, we train our method as well as a ProNet network on this dataset, and show enhanced results. For more detailed results, please refer to the general answer (point 3).
>
> >For the BindingSite Identification experiment, there is a missing baseline from dMaSIF which also achieves a 0.87 AUROC.
>
> There seems to be a minor confusion here, since the reported performance is done on the Masif-Ligand task that was not addressed in the dMasif paper. Validation on Masif-Site is also a potential extension of the validation of our work, but we chose Masif-Ligand instead as it was less redundant with PIP and AbAG tasks. Moreover, we included a Masif-Ligand-like task on our new validation on PINDER.
>
> >For most of these tasks, the dataset sizes available are typically limited. This makes it paramount to include uncertainty estimates, measured across 3 experimental runs. [...] Could the author(s) include the uncertainty estimates, measured across 3 experiment runs. I presume running these would not be out-of-scope for the rebuttal timeline, given the small size of the model and the datasets.
>
> We have included these results in the updated submission, and overall report very stable results across seeds. For more details, please see the general answer (point 1).
>
> >Presentation:
> The experiments section should include the corresponding table for the ribosomes (rather than in the appendix), or reorder the sections such that it falls under Ablation studies. This would improve the readability and flow of the paper.
>
> We agree that this would improve the readability of the paper. We will do our best to address this change, while adhering to the space constraints imposed by the conference template.
>
> >For the different benchmark tasks, it would be very beneficial to understand / detail how the training splits were constructed. A common practice is to construct splits based on sequence similarity, but these don't capture the structural nuances / differences which are more relevant for OOD generalization.
>
> We agree that data splitting is of utmost importance and added more details to the description of the data sets. Existing benchmarks almost all rely on sequence-based splits, but the Pinder validation does not. On this more stringent split, despite a moderate performance loss, our model is able to generalize and have a strong AuROC performance.
>
> &nbsp;
>
> We have tried to address all of the comments and questions by the reviewer, and thank the reviewer again for helping us improve our submission. We invite the reviewer to also consider the general feedback we provided in our general comment to all reviewers. As mentioned in our general comment, we will make sure to address *all* comments or concerns in the potential final version of our paper.

---

> > ### Comment · Reviewer_3ZDU · 2024-11-23
> > **Review Acknowledged -- Reviewer 3ZDU**
> >
> > I thank the authors for the time and effort spent on running the experiments for the rebuttal. The results on the harder PINDER split both on apo and holo structures (refer to general response) indeed highlights the improved performance of the method.
> >
> > > There seems to be a minor confusion here, since the reported performance is done on the Masif-Ligand task that was not addressed in the dMasif paper.
> >
> > That is correct, I overlooked this fact upon seeing the experiment titled BindingSite in both the dMaSIF and AtomSurf papers.
> >
> > All my concerns have now been addressed, and I am happy to increase my score. I think this is a nice paper that carefully tries to evaluate the utility of protein surfaces for representations, and I hope to see some of these ideas in the future used for generative models on surfaces.

---

> ### Author Response · Authors · 2024-11-25
> **Thank you**
>
> Thank you for your answer and for raising your evaluation of our work !

---

### Author Response · Authors · 2024-11-21
**General answer and additional results.**

We thank reviewers for their constructive comments and overall positive feedback. We are encouraged that the reviewers liked our idea of directly using and also combining surface-based learning (R1, R4), as well as adapting DiffusionNet for proteins (R3). They approved our experimental setup, empirical results and ablation studies and all appreciated the overall presentation of our paper. We appreciate the thoughtful suggestions for improvement and we will make sure to address *all* the comments, in the potential final version.

Several of the comments were consistent and we identified three major directions for improving our paper:
1. Assessing the reproducibility of our results across 3 seeds (R1, R3)
2. Assessing the impact of using ESM in the ablation study (R2, R4)
3. Expanding our benchmark tasks (R1, R3, R4)

We have addressed points 1 and 2 already, and are working on the implementation of point 3.

&nbsp;

1. Assessing the reproducibility of our results across 3 seeds

We have performed replicates of our method and overall report very stable performance, which confirms that our method outperforms competition on most benchmarks and validates its robustness. We present our results in the following table, where we show the performance of the best method excluding our contribution, the performance we reported in the submission and the mean performance and standard error obtained with three replicates.

|          Task         |    PIP   |    MSP   |    PSR   |          |   Ab   |        |   Ag   |        |  Masif Ligand |
|:---------------------:|:--------:|:--------:|:--------:|:--------:|:------:|:------:|:------:|:------:|:-------------:|
|         Metric        |   AuROC  |   AuROC  |   $R_l$  |   $R_g$  |  AuROC |   MCC  |  AuROC |   MCC  | Balanced acc. |
| _Runner-up method_    |   _87.1_ |   _68.5_ |   _63.2_ |   _86.8_ |   _80_ |   _28_ |   _72_ |   _18_ |          _81_ |
| Ours - Previous       |     90.7 | **71.5** |     61.3 | **86.1** |     86 |     47 | **73** | **18** |            88 |
| Ours - Mean           | **90.9** | **71.5** | **61.7** |     85.7 | **88** | **53** |   67.3 |   14.3 |      **88.2** |
| Ours - standard error |    0.088 |        1 |     0.27 |     0.22 |      1 |    2.1 |    1.8 |    1.5 |           0.3 |

In this table, we notice that our results are very stable. Indeed, our updated results show little difference compared to the difference across methods. As expected, the standard error is higher for smaller datasets (MSP, AbAg) than larger ones (PIP, PSR). Finally, we see that this variability affects the results on the AbAg task, where our performances are slightly enhanced on antibodies and decreased on antigens.
We added this table in the Supplemental.

&nbsp;

2. Assessing the impact of ESM ablation

We conducted an Ablation of our method by training models without the ESM embeddings as input features. The performance decreases as expected, showing that ESM is an important feature. However, our ablated method still shows very competitive results on most tasks, such as PIP, AbAg and MasifLigand.

|          Task         |    PIP   |    MSP   |    PSR   |          |   Ab   |        |   Ag   |        |  Masif Ligand |
|:---------------------:|:--------:|:--------:|:--------:|:--------:|:------:|:------:|:------:|:------:|:-------------:|
|         Metric        |   AuROC  |   AuROC  |   $R_l$  |   $R_g$  |  AuROC |   MCC  |  AuROC |   MCC  | Balanced acc. |
| Runner-up method      |     87.1 |     68.5 |     63.2 | **86.8** |     80 |     28 | **72** | **18** |            81 |
| Ours - Mean           | **90.9** | **71.5** | **61.7** |     85.7 | **88** | **53** |   67.3 |   14.3 |      **88.2** |
| Ours - no ESM         |       90 |     54.5 |       47 |       82 |     86 |     48 |     68 |   14.5 |            84 |

Please note that we did not tune our hyperparameters to adapt our training after this significant change. The overall difference was more noticeable on smaller datasets, especially on MSP for which the ablated network failed to learn a useful signal without ESM. Considering that AtomSurf-bench (which does not use ESM embeddings) gets 70.9 AuROC on MSP, we believe a high performance could be recovered by tuning the optimization. Based on this ablation, we conclude that ESM features are an important source of information, but that even without them, mixing a surface and a graph method gives a strong performance.
We now mention this result in our paper and have added this table in the Supplemental.

---

> ### Author Response · Authors · 2024-11-21
> **General answer and additional results (continued)**
>
> 3. Expanding our benchmark tasks
>
> Several reviewers have raised concerns about the tasks and datasets we used to benchmark our model. R1 has pointed us towards ProteinWorkshop, R2 to Pinder and Plinder and R4 mentioned doubts with regards to the relevance of some ATOM3D tasks.
>
> ATOM3D proposes a clear splitting strategy (30% seq identity) that we adhered to and which is more stringent than the one used in ProteinWorkshop, and the test performance in the PIP task is computed over apo structures.
>
> Moreover, we would like to emphasize that benchmarks more recent than ATOM3D might be better, but they lack adoption. We used only externally reported performance for other methods, and trained our methods sticking to their problem formulation and substituting just the protein structure encoder. We believe this is a fairer comparison than implementing other methods ourselves, since the amount of hyperparameter tuning influences the relative performances. Within this approach, new benchmarks do not offer the possibility to compare to externally reported performance. We have expanded the text of the relevant sections to put this vision forward.
>
> That being said, we also found the Pinder dataset highly interesting, due to its scale, diversity and more relevant data splitting based on interface similarity. We had started to work on setting up a training framework for that dataset before the rebuttal. The scale of the dataset makes it challenging to train on the whole of it using academic resources, but we were nevertheless able to train models on the clustered version.
>
> &nbsp;
>
> === UPDATE ===
> We have updated this block of results in the further comment
>
> We consider a task close to Masif-Site (mentioned by R1), only taking one protein as input to predict potential binding sites. We computed the performance of AtomSurf (ours) and Pronet and present results in the following table.
>
> |   Task   | Pinder Site |          |
> |:--------:|:-----------:|:--------:|
> |  Metric  |    Auroc    |    Acc   |
> |  Pronet  |        74.3 |     69.2 |
> | AtomSurf |    **88.3** | **82.3** |
>
> AtomSurf widely outperforms ProNet on both metrics. Moreover, we are able to obtain a high classification performance on this challenging dataset, indicating that our method is able to generalize.
>
> === END OF UPDATE ===
>
> &nbsp;
>
> We are still working on our submission and hope to offer more detailed results over the weekend. We are pushing on the following tasks:
> - Splitting the PINDER performance on the apo and predicted structures test set. This is not straightforward as the interfaces for those systems are not labeled in Pinder and present some alignment gaps to their bound counterparts.
> - Adding another PINDER task using the same formulation as PIP (detect interacting pairs of residue)
> - Computing an ESM-ablation of our results
> - Integrating our new results into the format constraints imposed by the conference template.
>
> &nbsp;
>
> We again thank reviewers for their overall positive feedback and for allowing us to improve our contribution. We hope those additional experiments alleviate some of the concerns you had expressed. We additionally provide individual answers to address the questions of each reviewer.

---

> ### Author Response · Authors · 2024-11-22
> **General answer and additional results (UPDATE)**
>
> We obtained novel results on the PINDER task, following our previous message. We post those as an update. Compared to what we had before (Pinder Site on Holo), we added another task formulated as the PIP task: given pairs of residues from two monomers, the network must classify them as interacting on non-interacting. Those pairs sampled to have a balanced distribution.
>
> Moreover, we added validation on the apo set and on the predicted set. The apo set is composed of monomers in an unbound set, representing a more realistic use case. The predicted set is composed of monomer structures predicted from sequence using AlphaFold2. Our results are presented in the following table.
>
> |                    |     Holo    |          |             |          |     Apo     |          |             |          | Predicted   |          |             |          |
> |--------------------|:-----------:|:--------:|:-----------:|:--------:|:-----------:|:--------:|:-----------:|:--------:|-------------|----------|-------------|----------|
> |        Task        | Pinder Pair |          | Pinder Site |          | Pinder Pair |          | Pinder Site |          | Pinder Pair |          | Pinder Site |          |
> |       Metric       |    Auroc    |    Acc   |    Auroc    |    Acc   |    Auroc    |    Acc   |    Auroc    |    Acc   |    Auroc    |    Acc   |    Auroc    |    Acc   |
> |       Pronet       |        80.1 |     72.5 |        74.3 |     69.2 |        78.2 |     71.5 |        70.7 |       71 |        73.5 |     67.9 |        60.6 |     56.3 |
> |      AtomSurf      |        92.8 |     85.3 |    **88.3** | **82.3** |    **88.4** | **80.7** |    **84.2** | **80.8** |    **87.1** | **80.2** |      **82** |     76.6 |
> | AtomSurf -  no ESM |      **93** | **85.4** |        87.6 |     79.2 |        87.6 |     80.2 |        82.4 |     77.1 |          87 |     79.9 |        80.9 | **78.8** |
>
> The first thing to point out is that we widely outperform the ProNet baseline on all tasks, splits and metrics. Moreover, as expected, the performance over the apo set is decreased compared to the holo set, as well as the performance on predicted structures, a well-known result reported for instance in [1]. We can compare the performance on the Apo set to the one obtained on PIP (that also uses apo structures). The PINDER test set is more challenging because it contains much fewer dimers (which are easier to predict) and because the split is more challenging. However, our network is able to retain a remarkable accuracy across the board, validating its robustness.
>
> We also included an ablation of the ESM features, in line with point 2 above. While we still obtain a small benefit from using those features, most notably on the apo set, this gap seems to be quite small on this dataset with stricter splits. One possible explanation for this result would be that those sequence-derived features contribute to memorization, whilst learnt structural properties could better generalize.
>
> We hope those additional results will be as exciting to the reviewers as they are to us. We will now focus on blending those results in our initial submission.
>
> [1] Protein 3D Graph Structure Learning for Robust Structure-based Protein Property Prediction. Huang et al. AAAI 2024

---

### Meta-Review · Area_Chair_Dg5r · 2024-12-22

**Metareview:**

The paper received positive support from most of the reviewers, and the overall recommendation was positive. Thus an accept is recommended.

**Additional Comments On Reviewer Discussion:**

Most of the issues have been resolved, and the reviewer giving a score of 5 did not respond to author rebuttals.

---

### Decision · Program_Chairs · 2025-01-22

Accept (Poster)